# NATURE-INSPIRED POPULATION-BASED EVOLUTION OF LARGE LANGUAGE MODELS

## ABSTRACT

Evolution, the engine behind the survival and growth of life on Earth, operates through the population-based process of reproduction. Inspired by this principle, this paper formally defines a newly emerging problem: the population-based evolution of large language models (LLMs). We introduce a novel framework that starts with a population of parent LLMs and allows this population to evolve through four key operations: (i) crossover, merging the weights of different parents to create offspring LLMs, (ii) mutation, introducing small, random changes to model weights to foster diversity, (iii) selection, prioritizing high-performing models, and (iv) succession, transferring the learned experience from parent to offspring LLMs. With only 200 samples per new task, the LLM population evolves rapidly to adapt to the task at hand, without any gradients. Experiments on 12 datasets show that our framework consistently outperforms existing multi-LLM merging and adaptation methods, achieving relative accuracy gains of up to 54.8% over the best LLM in the initial population. Moreover, our framework allows for (i) the evolution of LLMs across multiple new tasks simultaneously, (ii) scaling effectively with populations of up to 40 LLMs, and (iii) even zero-shot generalization to unseen held-out tasks.

## 1 INTRODUCTION

For billions of years, nature has taught us a profound lesson: *evolution* drives the survival and flourishing of all life on Earth, from the simplest single-celled organisms to complex ecosystems and human civilizations. Evolution works not on individuals in isolation but on entire populations, where *"survival of the fittest"* shape the story of life. Now, imagine channeling these very principles not to biological species but to cutting-edge AI technology—what if we could harness population-based evolution to advance large language models (LLMs)?

There are multiple reasons for the population-based evolution of LLMs to be attractive. First, there is a growing proliferation of fine-tuned "expert" models that are specialized to specific tasks; for example, there are more than 170,000 expert models, trained through parameter-efficient fine-tuning techniques, available on Hugging Face (Muqeeth et al., 2024), which lies a solid foundation for conceptualization an LLM population. Second, recent studies have shown that merging the weights of LLMs with different expertise can produce versatile models (Yu et al., 2024; Goddard et al., 2024; Mavromatis et al., 2024), sometimes even resulting in emergence of new capabilities (Yadav et al., 2024). Third, enabling the evolution of an LLM population could allow these models to dynamically adapt to new, unseen tasks that have not been explicitly trained on (Huang et al., 2024; Mavromatis et al., 2024). This effectively recycles the substantial efforts and computing resources invested in training existing models. Last but not least, it also open the door to democratizing foundation model development (Akiba et al., 2024), encouraging a broader community to jointly advance the field.

In this paper, we formalize the LLM population-based evolution problem, and propose a novel solution inspired by the very basic units of life—genes. Just as genes define biological traits, LLM weights define a model's capabilities and behaviors. This naturally leads us to the genetic algorithm (GA) (Holland, 1992), a well-established evolutionary algorithm for optimization. We generalize GA to solve the LLM population-based evolution problem, and introduce **GENOME** (**GEN**etic **O**ptimization for **M**odel **E**volution). GENOME adapts the key concepts of GA as follows: (i) **fitness** measures how well a LLM performs (or fits) on a specific task, (ii) **crossover** merges the

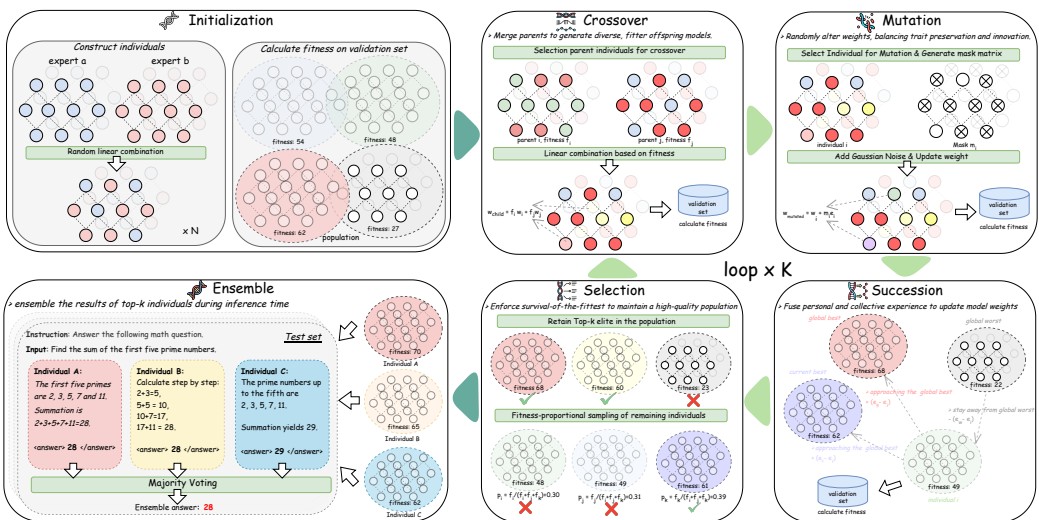

Figure 1: **GENOME+: an easily accessible, gradient-free solution to the LLM population-based evolution problem.** We generalize the well-established genetic algorithms to solve the LLM population-based evolution problem, and additionally introduce two new operators. Starting with a pool of task-specific expert LLMs, the *fitness* of each model is evaluated based on its performance on a validation set (comprising only 200 samples). Through *crossover*, we combine the parameter weights of different parent LLMs to generate offspring LLMs. *Mutation* introduces small random perturbations to inject controlled variability, while *selection* retains top-performing LLMs, ensuring population diversity. *Succession* allows LLMs to learn from the best-performing models, and *ensemble* aggregates answers within the population during inference. This process repeats for multiple generations, enabling gradient-free adaptation and consistent improvements across tasks.

weights of different LLMs, generating offspring models and enabling them to inherit advantageous features from parent models, (iii) **mutation** adds small random perturbations to the model weights, enabling the model to acquire new abilities, and **selection** selects high-performing models to survive, effectively driving the LLM population to evolve in a better direction.

While connecting GA to LLMs is conceptually appealing, we recognize that GA heavily relies on reproduction, overlooking broader evolutionary principles, such as learning from peers and group decision-making. Thus, we additionally introduce two operations: (i) **succession**, which allows LLMs to learn from the best-performing LLM in the population, and (ii) **ensemble**, which integrates the outputs of top-$k$ best performing models into a final answer. This results in **GENOME+** (as shown in Figure 1). Compared to GENOME, GENOME+ makes better use of the diversity within a LLM population. Despite their differences, both methods (collectively referred to as **GENOME(+)**) are easy to implement. Moreover, they allow for gradient-free weight updates, eliminating the need for backpropagation process. This provides an easily accessible solution to the LLM population-based evolution problem.

We conduct extensive evaluations on 12 datasets, showing that GENOME(+) consistently outperforms 7 methods, with only 200 samples per dataset. Unless otherwise stated, all percentage gains reported below are relative improvements in accuracy over the corresponding baseline. Notably, compared to the state-of-the-art Model Swarms (Feng et al., 2024), GENOME+ averages +10.75% and excels in reasoning-intensive tasks, improving by 36.3% on DROP (Dua et al., 2019) and 19.3% on MGSM (Shi et al., 2022). When evolving to simultaneously adapt to two tasks, GENOME+ surpass Model Swarms by 13.46% on average, 50.9% on DROP, 17.6% on EmoryNLP (Zahiri & Choi, 2018), and 16% on GSM8k (Cobbe et al., 2021). Furthermore, GENOME+ generalize zero-shot across tasks, achieving an average +11.72% cross-dataset gain over *Expert Fusion* (other baselines are ∼0–1% or negative). On unseen languages, GENOME/GENOME+ improve over the *Best Single* by +12.83%/+15.98% on Flores-101 (Goyal et al., 2022). Even as the population scales up to 40 LLMs, both methods continue to demonstrate performance improvements. To assess computational efficiency, we repeat our key experiments on a single 4090 GPU with 24GB of memory. The results show that our method maintains consistent performance without requiring extensive compu-

tational resources. Last but not least, an ablation study confirms the positive contribution of each evolutionary operation.

We have open-sourced our code on GitHub: `https://anonymous.4open.science/r/GENOME-0DBD`. To summarize, our key contributions are as follows:

- We formally define the LLM population-based evolution problem and present a novel conceptual framework that connects LLM weights to genes, task performance to individual fitness, and weight updates to evolutionary operations.
- We propose GENOME and GENOME+, which incorporate the key principles of traditional genetic algorithms, namely, crossover, selection, and mutation, as well as two operations: succession and ensemble. This showcases how traditional evolutionary algorithms can shed light on the population-based evolution of LLMs.
- We conduct extensive experiments in single task, multi-task domain, and zero-shot generalization settings, demonstrating GENOME+ 's advantages, verifying its scalability to 40-model populations, and confirming each operation's necessity through ablation studies. Experiments show that our results can be reproduced on a 4090 GPU with 24GB of memory.

## 2 RELATED WORK

**Model Parameter Merging.** Leveraging the complementary strengths of multiple LLMs has recently attracted much interest (Zhao et al., 2024; Wang et al., 2024a; Chen et al., 2024). Within this landscape, our work is most closely related to the line of research on model parameter merging. Earlier approaches, such as, Model Soup (Wortsman et al., 2022), Model Stock (Jang et al., 2024), TIES (Yadav et al., 2023), and DARE (Yu et al., 2024), typically merge the parameters of multiple models into those of a single one based on predefined rules, and maintain the model parameters *unchanged* after merging. More recent studies have extended to explore *dynamic* parameter merging for adapting to new tasks (Hu et al., 2024; Akiba et al., 2024; Yang et al., 2024; Huang et al., 2024). Akiba et al. (2024) propose evolving the mixing weights at each layer within the parameter space, as well as evolving layer permutations within the data flow space. LoRAHub (Huang et al., 2024) formulates dynamic parameter merging as a linear-combination problem and optimizes linear coefficients. MERGE[3] (Mencattini et al., 2025) similarly applies multi-objective evolutionary search over full-parameter mixing coefficients, but is instantiated in a full-weight setting rather than the LoRA-based multi-domain regime. Likewise, PackLLM (Mavromatis et al., 2024) introduces a test-time merging strategy for combining by solving an optimization problem to minimize perplexity without requiring training. Model Kinship (Hu et al., 2024) assesses relatedness between LLMs, guiding merges of top-performing models with those of distinct task capabilities. Model Swarms (Feng et al., 2024) conceptually treats each LLM as a particle and adapts particle-swarm optimization to iteratively identify the best "particle" for a given new task.

Compared to these studies, our work bears several important differences. First, instead of producing a single final model as in the aforementioned studies, we formulate and address the population-based evolution of LLMs in an adapter-based (LoRA) setting, maintaining and evolving a diverse set of models over time. Second, technically, we instantiate this view by letting GA-style operations—including but not limited to mutation, crossover, selection, and succession—act directly on LoRA adapters, rather than on low-dimensional mixing coefficients between a fixed set of models, and we find that this design consistently improves over representative static and dynamic parameter merging methods across a wide range of tasks.

**Evolutionary Algorithm.** Our study also relates to research that explores the intersection of evolutionary algorithms and LLMs (see a recent review (Wu et al., 2024) and references therein). However, unlike our approach, prior work in this area has primarily focused on two directions: (i) using LLMs as evolutionary operators to heuristically guide the search in traditional optimization problems (Liu et al., 2024; van Stein & Bäck, 2024), and (ii) employing evolutionary algorithms to iteratively optimize prompts (Guo et al., 2023; Baumann & Kramer, 2024) or agent workflows (Yuan et al., 2024; Jin et al., 2024). In contrast, our work demonstrates that a classic evolutionary algorithm (i.e. GA) can be generalized to optimize the model parameters themselves. This complements existing research and further broadens the application of evolutionary principles in LLM development.

**Self-Evolving LLM or LLM agents.** It worth mentioning that, recently, there is growing interest in the evolution of LLM or LLM agents (Gao et al., 2025). Most studies in this area focus on the evolution of a single model or agent, with methods ranging from reinforcement learning (Zhao et al., 2025; Wang et al., 2025b) and prompt optimization (Fernando et al., 2023) to memory management (Wang et al., 2025a). In contrast, our work is the first to formally define the LLM population-based evolution problem and one of the first to demonstrate how evolutionary algorithms can be generalized to evolve a population of models.

# 3 METHODOLOGY

## 3.1 LLM POPULATION-BASED EVOLUTION

We define the **LLM population-based evolution** as an iterative optimization problem over a population of large language models (LLMs). Given a population: $P^{(t)} = \{x_i^{(t)}\}_{i=1}^N$, $x_i^{(t)} \leftrightarrow \mathbf{w}_i^{(t)} \in \mathbb{R}^d$, where each model $x_i^{(t)}$ is represented by learnable parameters $\mathbf{w}_i^{(t)}$, which may encompass either the full LLM weights or a subset thereof (such as LoRA or adapter parameters). The performance of each individual model on a given task $T$ is quantified by a function $f(\mathbf{w}_i^{(t)}; T)$. The goal is to evolve the population $P^{(t)}$ over iterations $t = 0, 1, \ldots, K$ to optimize the aggregated fitness of the top-$k$ individuals in the final population $P^{(K)}$:

$$\max_{\{P^{(t)}\}_{t=0}^K} \frac{1}{k} \sum_{i=1}^k f(\mathbf{w}_{(i)}^{(K)}; T) \tag{1}$$

where $\mathbf{w}_{(i)}^{(K)}$ denotes the parameters of the $i$-th best-performing individual in $P^{(K)}$ ranked by fitness. The integer $k \in [1, N]$ controls whether the optimization focuses solely on the best individual ($k = 1$), or collectively improves a larger subset or the entire population ($k > 1$).

## 3.2 GENOME

In nature, biological evolution operates through an iterative process of **crossover**, **mutation**, and natural **selection**, where genetic variation drives diversity and selection preserves traits with higher adaptability. This cycle enables species to refine survival strategies in dynamic environments. Viewed through this lens, LLM performance reveals similar dynamics: each model represents an individual with unique traits, its weight parameters forming a digital genome, and task performance informs the selection mechanism that governs its chances of "survival" and "reproduction".

We propose GENOME (**GEN**etic **O**ptimization for **M**odel **E**volution). Here, the target task $T$ serves as the "environment", and the **fitness function** $f$ is the performance on the validation set of $T$. Starting with a population of $n$ homogeneous LLM expert models (derived from the same base model), we propose to create an initial population $P^{(0)} = \{x_i\}_{i=1}^N$ via an **initialization** operation, where each individual's genes are represented by its **LoRA** weights $\mathbf{w}_i$. The **crossover** operation is employed to creates new offspring LLMs through merging parameters from parent LLMs, while the **mutation** operation is utilized to introduce variations to the parameters. The **selection** operation simulates natural selection pressure by retaining advantageous individuals. Algorithm 1 outlines GENOME, which includes initialization, crossover, mutation, and selection, executed over $K$ iterations.

**Initialization** Starting with $n$ LLM expert models $\{x_i\}_i^n$, we construct a diverse initial population through random linear combinations of expert model weights. For each new individual $i$, we combine the LoRA parameters $\mathbf{w}_a$ and $\mathbf{w}_b$ from two expert models:

$$\mathbf{w}_i \leftarrow t \cdot \mathbf{w}_a + (1-t) \cdot \mathbf{w}_b, \quad t \sim U(0,1) \tag{2}$$

This process is repeated $N$ times to form a population of size $N$. We then evaluate each individual on a validation set using the fitness function to obtain fitness scores $f_i$ for the evolution process.

**Crossover** During sexual reproduction, chromosomal crossover creates novel genetic combinations that increase population diversity, providing raw material for natural selection to enhance

adaptability. GENOME draws on this mechanism by combining the weights of different models. Specifically, we first design a selection probability based on individual fitness: $p_i = f_i / \sum f_k$, where $k = 1, 2, \ldots, N$. Using this probability, we select parent pairs $(x_{p_1}, x_{p_2})$ with their corresponding weights $(\mathbf{w}_{p_1}, \mathbf{w}_{p_2})$. The offspring is generated by combining these weights:

$$\mathbf{w}_{child} \leftarrow t \cdot \mathbf{w}_{p_1} + (1 - t) \cdot \mathbf{w}_{p_2} \tag{3}$$

where $t = f_{p_1} / (f_{p_1} + f_{p_2})$ is the normalized weight. The crossover rate $(c_r)$ controls the proportion of population undergoing this process, preserving model diversity while improving performance.

**Mutation** In biological evolution, gene mutations introduce random variations that maintain population diversity. Following this principle, GENOME applies mutation in two stages. First, each individual $x_i$ is selected for mutation with probability $im_r$. Then, for each selected model, a binary mask $\mathbf{m}_i$ is generated with probability $gm_r$ to determine which weight parameters will be mutated. The mutation is executed as follows:

$$\mathbf{w}_i' \leftarrow \mathbf{w}_i + \mathbf{m}_i \cdot \mathbf{E}_i, \quad \mathbf{E} \sim \mathcal{N}(0, \sigma^2) \tag{4}$$

where the binary mask $\mathbf{m}_i$ identifies the parameters to change, while the Gaussian noise $\mathbf{E}_i$ and its standard deviation $\sigma$ control the mutation intensity. The resulting mutated weights $\mathbf{w}_i'$ form new individuals $x'$ in the population, balancing the preservation of successful traits with the introduction of innovative variations.

**Selection** Selection in GENOME serves to simulate natural selection pressure while maintaining a fixed population size $N$. After crossover and mutation operations, the selection process first retains the top $\alpha N$ individuals with the highest fitness scores ($\alpha \in (0, 1)$). Then, to restore the population size to $N$, individuals are selected from the entire current population with probabilities proportional to their fitness scores, where the selection probability for individual $i$ is calculated as $p_i = f_i / \sum_{k=1}^{N} f_k$. This process effectively maintains population stability while favoring better-performing individuals.

### 3.3 GENOME+

While GA traditionally emphasize evolution at the genetic level, the evolutionary history of intelligent species highlights the importance of knowledge transfer and collective decision-making within social groups. When models are viewed as a population, their evolution should involve not only parameter crossover and mutation, but also mechanisms for experience sharing and group decision-making. Motivated by this, we introduce GENOME+, which adds two new operations to the GENOME to achieve these capabilities. Through **succession**, individuals inherit successful strategies from the population's evolutionary history, while avoiding prior failures. **Ensemble** methods combine the outputs of diverse models, leveraging their collective intelligence. Algorithm 2 outlines GENOME+.

**Succession** Model Swarms (Feng et al., 2024) demonstrates that Particle Swarm Optimization (PSO) can effectively optimize LLM groups by tracking personal best, global best, and global worst solutions. Building on this insight, we represent each individual's learning pattern as an experience vector $\mathbf{e}_i$, which is randomly initialized with the same dimensionality as the LoRA weights $\mathbf{w}_i$, and facilitate knowledge transfer within the population through experience updates. The experience update for each individual integrates four sources: global best $\mathbf{e}_g$ (the best experience in the population's evolution so far), current best $\mathbf{e}_c$ (the latest successful pattern adapting to dynamic environmental changes), global worst $\mathbf{e}_w$ (failed experience as a negative example), and self-experience $\mathbf{e}_i$ (the individual's accumulated experience). The update follows:

$$\mathbf{e}_i \leftarrow \frac{1}{\mathbf{C}} [\phi_e \mathbf{e}_i + \phi_g (\mathbf{e}_g - \mathbf{e}_i) + \phi_c (\mathbf{e}_c - \mathbf{e}_i) - \phi_w (\mathbf{e}_w - \mathbf{e}_i)] \tag{5}$$

where $\phi_*$ are the weights for each source of experience, and $\mathbf{C} = \phi_e + \phi_g + \phi_c + \phi_w$ is the normalization coefficient. The updated experience is then applied to adjust the model weights: $\mathbf{w}_i \leftarrow \mathbf{w}_i + \lambda \mathbf{e}_i$, where $\lambda$ is the learning rate for experience. This mechanism allows individuals to maintain their unique characteristics while also leveraging the collective wisdom during the evolutionary process.

**Ensemble** GENOME+ introduces an ensemble operation during the inference phase, selecting the top-k individuals with the highest fitness from the evolved population and aggregating their prediction results. This ensemble decision-making mechanism integrate the outputs of multiple high fitness individuals, while fully leveraging the advantages accumulated by the population during the evolutionary process. Through result aggregation in the inference phase, GENOME+ can achieve more robust predictive performance.

# 4 EXPERIMENTS

In the following discussion, **GENOME(+)** refers to GENOME and GENOME+ together. Our experiments investigate the following key questions: (i) How effectively GENOME(+) adapt to a single task compared to existing methods? (ii) Can the framework maintain stable performance when handling multiple tasks simultaneously? (iii) Can the framework generalize to unseen, held-out tasks? (iv) How well does our framework scale with increasing population size? (v) How does hardware configuration affect time efficiency and overall performance? (vi) What is the impact of each operator on overall performance? The complete codebase used in our experiments is publicly available on `https://anonymous.4open.science/r/GENOME-0DBD`.

## 4.1 EXPERIMENTAL SETUP

**Models** We employ *gemma-2-2b-it* (Team et al., 2024) as our foundation model, and construct a set of domain-specific experts by fine-tuning the foundation model on 10 distinct domains extracted from the **Tulu-v2-SFT-mixture** dataset (Ivison et al., 2023b). The fine-tuning process is implemented using the llama-factory framework (Zheng et al., 2024), incorporating the low-rank adaptation (LoRA) technique (Hu et al., 2021). We further demonstrate the capabilities of these 10 experts across different domains (see Figure 3), which confirms that our training process yields expert models with varying proficiencies. Complete training hyperparameters and configurations are detailed in Table 8.

**Datasets** We consider 12 datasets covering 7 key capabilities of LLMs, including **General Knowledge** (MMLU (Hendrycks et al., 2021a), MMLUPro (Wang et al., 2024b)), **Mathematics** (MATH (Hendrycks et al., 2021b), GSM8K (Cobbe et al., 2021), and MGSM (Shi et al., 2022)), **Code Generation** (MBPP (Austin et al., 2021)), **Logical Reasoning** (DROP (Dua et al., 2019), BBH (Suzgun et al., 2022)), **Multilingual Processing** (MGSM (Shi et al., 2022) and FLORES-101 (Goyal et al., 2022)), **Affective Computing** (EmoryNLP (Zahiri & Choi, 2018), MELD (Poria et al., 2018)), and **Question Answering** (ARC_C (Clark et al., 2018) and CSQA (Talmor et al., 2019)). Note that MGSM is a multilingual extension of mathematics problems, thus appearing in both Mathematics and Multilingual Processing. Each dataset is divided into a 200-sample validation set and a test set comprising approximately 1,000 samples. Detailed descriptions of the data splits and metrics can be found in Appendix A.2 and Table 7. Notably, these datasets do not overlap with the training data used for the expert models.

**Implementation of GENOME(+)** Unless otherwise stated, we use the following setting: crossover rate ($c_r$) of 0.3, individual mutation rate ($im_r$) of 0.3, gene mutation rate ($gm_r$) of 0.2, and standard deviation ($\sigma$) of 0.001. Additionally, the maximum number of iterations is set to 10, the population size is 10, $\phi_*$ is (0.95, 0.2, 0.2, 0.1), $\lambda$ is 0.95. For ensemble operation, we select the top-3 individuals with the highest fitness scores on the validation set.

**Baselines** We compare GENOME(+) to 7 baselines:

- Best Single, which refers to the best-performing expert model for a given dataset.
- Data Merge, which combines the 10 sub-datasets of Tulu-v2-sft-mixture into one complete training dataset for training a single LoRA model.
- Expert Fusion, which merges the 10 expert models through $\mathbf{w}_{\text{fusion}} = \sum_{i=1}^{n} t_i \mathbf{w}_i$, where $t_i$ denotes the normalized fitness value of each expert model $i$.
- DARE_TIES (Yu et al., 2024), which merges multiple LoRA models into a single one, while effectively reducing parameter interference.

Table 1: Performance comparison of different methods across 12 datasets (averaged over 5 runs with different random seeds). Best results are in bold and second-best results are underlined. In comparison rows, red/green percentages indicate improvements/decreases respectively, and '—' denotes no change. GENOME+ achieves average improvements of **24.06%/10.75%** over Best Single/Model Swarms respectively (max **54.80%/36.33%** on DROP).

| Method | MMLU | MMLUPro | GSM8k | MATH | MGSM | Flores-37 | ARC_C | CSQA | BBH | DROP | EmoryNLP | MBPP |
|---|---|---|---|---|---|---|---|---|---|---|---|---|
| Best Single | 52.90 | 26.57 | 40.80 | 14.30 | 30.22 | 21.51 | 57.34 | 64.30 | 30.22 | 30.40 | 32.53 | 31.78 |
| Data Merge | 13.60 | 25.57 | 33.00 | 12.20 | 25.98 | 22.51 | 46.08 | 37.40 | 25.98 | 27.40 | 32.37 | 15.37 |
| DARE_TIES | 48.30 | 28.57 | 35.50 | 12.50 | 31.85 | 22.33 | 69.80 | 23.90 | 31.85 | 19.22 | 34.22 | 37.77 |
| Expert Fusion | 55.80 | 27.67 | 39.50 | 13.10 | 31.82 | 21.74 | 69.03 | 65.00 | 31.82 | 22.60 | 32.66 | 28.55 |
| LoraHub | 53.00 | 27.17 | 46.47 | 14.90 | 34.70 | 21.83 | 69.97 | 66.10 | 39.30 | 35.32 | 31.53 | 42.63 |
| MERGE³-LoRA | 53.16 | 28.09 | 48.03 | 14.90 | 37.81 | 22.19 | 70.01 | 67.68 | 39.40 | 33.28 | 32.77 | 42.48 |
| Pack of LLMs | 54.30 | 27.67 | 38.13 | 11.50 | 31.06 | 19.24 | 70.22 | 63.30 | 38.90 | 21.30 | 30.46 | 42.56 |
| Model Swarms | 55.91 | 27.77 | 45.82 | 15.06 | 33.15 | 21.76 | 68.53 | 68.76 | 38.80 | 34.52 | 33.98 | 42.56 |
| GENOME | 56.66 | 27.77 | 49.34 | 15.72 | 37.19 | 22.78 | 70.69 | 68.22 | 40.00 | 35.28 | 34.97 | 43.60 |
| - vs Best Single | +7.11% | +4.52% | +20.93% | +9.93% | +23.06% | +5.90% | +23.29% | +6.10% | +32.36% | +16.05% | +7.50% | +37.19% |
| - vs Model Swarms | +1.34% | – | +7.68% | +4.38% | +12.19% | +4.69% | +3.15% | -0.79% | +3.09% | +2.20% | +2.91% | +2.44% |
| GENOME+ | 56.44 | 30.98 | 51.24 | 16.41 | 39.55 | 23.38 | 74.38 | 69.89 | 41.10 | 47.06 | 38.81 | 43.54 |
| - vs Best Single | +6.69% | +16.60% | +25.59% | +14.76% | +30.87% | +8.69% | +29.72% | +8.69% | +36.00% | +54.80% | +19.31% | +37.00% |
| - vs Model Swarms | +0.95% | +11.56% | +11.83% | +8.96% | +19.31% | +7.44% | +8.54% | +1.64% | +5.93% | +36.33% | +14.21% | +2.30% |

- MERGE³-LoRA (Mencattini et al., 2025), originally designed for full-parameter model merging; in our experiments, we use a straightforward LoRA-based adaptation that applies the same merging procedure to the 10 LoRA experts on top of a shared frozen base model.
- LoraHub (Huang et al., 2024), which dynamically selecting and merging LoRA models, uses GA to optimize the combination weights.
- Pack of LLMs (Mavromatis et al., 2024), which dynamically merges LoRA models, assigning weights based on each model's perplexity for given input prompts.
- Model Swarms (Feng et al., 2024), which employs PSO to iteratively optimize the set of expert models and dynamically adapt to a given dataset.

Note that Expert Fusion, LoraHub, Pack of LLMs, Model Swarms, and our framework all require a few samples for dynamic adaptation (*dynamic methods*); for a fair comparison, we fix the number of samples to 200. Additionally, all aforementioned methods—except for Data Merge, which does not require multiple expert models—utilize the same set of 10 expert models trained on the Tulu-v2-SFT-mixture. The implementation details of these methods are illustrated in Appendix A.3.

## 4.2 RESULTS AND ANALYSIS

All experiments are conducted using five random seeds, and the reported results are averaged.

### 4.2.1 SINGLE TASK

Table 1 demonstrates that the GENOME and GENOME+ have attained superior performance across **12 datasets**. Specifically, GENOME+ exhibits the most robust performance—on average, enhancing by 24.06% relative to Best Single and by 10.75% in comparison to Model Swarms, thoroughly substantiating its efficacy. Our frameworks have demonstrated consistent performance enhancements across multiple categories, particularly in tasks necessitating intricate reasoning skills, such as BBH and DROP, where the framework attain increases of 36% and 54.80%, respectively. In Mathematics, it increases by 23.74% relative to the Best Single and by 13.37% in comparison to Model Swarms.

Nonetheless, we notice that in Question Answering (ARC_C, CSQA) and General Knowledge (MMLU, MMLUPro), the enhancements in performance are less significant. To further investigate this, we specifically analyze and compare performance on reasoning-intensive versus knowledge-intensive task subsets, where we observe a more pronounced improvement on reasoning tasks; detailed analysis result are provided in Appendix A.5 and Table 11.

### 4.2.2 MULTI-TASK DOMAIN

In our second experiment, we consider five domains each consists of two tasks. To allow for simultaneous adapation to two task,s we consider the fitness value to be the average of the two tasks'

performance. We only compare dynamic methods because the performance of static methods remains constant across tasks. Compared to single task, this scenario is more challenging.

As shown in Table 2, GENOME and GENOME+ are still able to achieve performance at the level comparable to single-task scenario, whereas other dynamic methods suffer from various degrees of performance degradation. Interestingly, GENOME+ achieves the best results in every category, proving the stability of our framework in navigating challenging situations.

Table 2: The results of multi-task domain.

| Method | Affective Computing | | Mathematics | | General Knowledge | | Question Answering | | Logical Reasoning | |
| | MELD | EmoryNLP | GSM8k | MATH | MMLU | MMLUPro | ARC_C | CSQA | DROP | BBH |
|---|---|---|---|---|---|---|---|---|---|---|
| Expert Fusion | 50.35 | 31.91 | 41.77 | 12.60 | 53.10 | 25.97 | 68.94 | 63.00 | 22.10 | 38.30 |
| LoraHub | 50.77 | 32.67 | 41.77 | 10.30 | 52.90 | 26.37 | 69.37 | 62.40 | 34.10 | 38.30 |
| Pack of LLMs | 51.70 | 29.81 | 37.90 | 11.90 | 53.40 | 27.07 | 69.03 | 63.10 | 21.20 | 38.60 |
| Model Swarms | 52.31 | 33.53 | 43.84 | 15.10 | 55.50 | 27.99 | 67.76 | 65.78 | 31.80 | 38.40 |
| GENOME | 52.68 | 35.08 | 46.64 | 14.48 | 55.52 | 28.31 | 69.73 | 67.92 | 37.90 | 39.15 |
| - vs Model Swarms | +0.71% | +4.62% | +6.39% | -4.11% | +0.04% | +1.14% | +2.91% | +3.25% | +19.18% | +1.95% |
| GENOME+ | 56.05 | 39.42 | 50.87 | 17.24 | 56.40 | 30.13 | 73.60 | 70.35 | 48.00 | 39.90 |
| - vs Model Swarms | +7.15% | +17.57% | +16.04% | +14.17% | +1.62% | +7.65% | +8.62% | +6.95% | +50.94% | +3.91% |

Table 3: Generalization performance across different domains.

| Method | MMLUPro → MMLU | MATH → GSM8k | EmoryNLP → MELD | Macro Avg. %Δ vs Expert Fusion |
|---|---|---|---|---|
| Expert Fusion | 53.40 | 42.53 | 50.75 | – |
| LoraHub | 53.50 | 43.37 | 49.92 | +0.18% |
| Pack of LLMs | 52.70 | 37.83 | 51.93 | -3.35% |
| Model Swarms | 52.10 | 44.05 | 51.29 | +0.74% |
| GENOME | 55.40 | 46.32 | 52.07 | +5.09% |
| GENOME+ | 56.65 | 51.10 | 55.28 | +11.72% |

Table 4: Performance on seen (**Flores-37**) and unseen languages (**Flores-101**).

| Method | Flores-37 | Flores-101 |
|---|---|---|
| Best Single | 21.51 | 12.70 |
| Expert Fusion | 21.74 (+1.07%) | 13.74 (+8.19%) |
| LoraHub | 21.83 (+1.49%) | 13.82 (+8.82%) |
| Pack of LLMs | 19.24 (-10.55%) | 12.40 (-2.36%) |
| Model Swarms | 21.76 (+1.16%) | 13.64 (+7.40%) |
| GENOME | 22.78 (+5.90%) | 14.33 (+12.83%) |
| GENOME+ | 23.38 (+8.69%) | 14.73 (+15.98%) |

### 4.2.3 ZERO-SHOT GENERALIZATION

We assess the zero-shot generalization capability through two experiments: cross-dataset transfer and unseen language adaptation. We identify three groups of related task pairs for cross-dataset transfer: MMLUPro to MMLU, MATH to GSM8k, and EmoryNLP to MELD. For each pair, we initially enable a dynamic method to adapt to one task with 200 samples, and then test its performance on the other task without any samples. Table 3 shows that all dynamic methods transfer, with GENOME/GENOME+ strongest; GENOME+ achieves a macro gain of **+11.72%** over Expert Fusion (vs. GENOME **+5.09%**), while other baselines are ≤ **+0.74%** or negative.

Additionally, we assess our framework using the Flores101 dataset, comprising 64 languages not included in the training phase of gemma-2[1]. Table 4 indicates that the majority of methods demonstrate performance improvements, with the exception of the Pack of LLMs. GENOME and GENOME+ demonstrate significant enhancements of 12.83% and 15.98%, respectively, exceeding their performance improvements on Flores37, which includes languages encountered during pre-training. The results indicate a robust generalization capability of our framework, especially in the context of low-resource languages.

### 4.2.4 SCALABILITY

We conduct comparison experiments using Model Swarms, GENOME, and GENOME+ to examine their scalability, increasing the population size from 10 to 40 by 10. In the MMLUpro, MATH, and MMLUPro$_{Reasoning}$ tasks, all approaches exhibit enhanced performance with an increase in population size, with GENOME+ attaining the highest performance across all sizes. As shown in Figure 2, GENOME consistently outperforms Model Swarms, while GENOME+ demonstrates the strongest performance. These results confirm that GENOME+ can scale to larger population sizes up to 40.

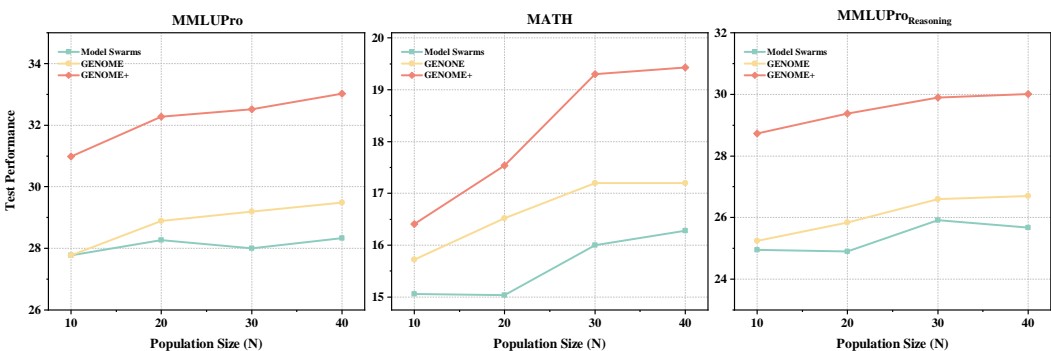

Figure 2: Performance trends with increasing population sizes (*N*) across different methods.

Table 5: Ablation study of different operations

| Setting | MMLUPro | GSM8k | DROP | BBH | EmoryNLP | MGSM |
|---------|---------|-------|------|-----|----------|------|
| **GENOME+** | **30.98** | **51.24** | **47.06** | **41.10** | **38.81** | **39.55** |
| *w/o initialization* | 29.72 (-4.07%) | 46.02 (-10.19%) | 42.70 (-9.26%) | 39.65 (-3.53%) | 38.29 (-1.34%) | 36.61 (-7.43%) |
| *w/o crossover* | 30.32 (-2.13%) | 48.86 (-4.64%) | 45.10 (-4.16%) | 40.00 (-2.68%) | 38.01 (-2.06%) | 35.72 (-9.68%) |
| *w/o mutation* | 29.72 (-4.07%) | 49.51 (-3.38%) | 43.90 (-6.71%) | 40.10 (-2.43%) | 37.17 (-4.23%) | 38.41 (-2.88%) |
| *w/o cro. & mut.* | 27.87 (-10.04%) | 37.00 (-9.98%) | 38.60 (-17.98%) | 35.29 (-9.07%) | 33.11 (-16.28%) | 45.38 (-11.44%) |
| *random selection* | 26.77 (-13.59%) | 44.96 (-12.26%) | 38.50 (-18.19%) | 38.91 (-5.33%) | 35.21 (-9.28%) | 37.67 (-4.75%) |
| *w/o succession* | 28.76 (-7.17%) | 50.15 (-2.13%) | 40.22 (-14.53%) | 39.72 (-3.36%) | 36.52 (-5.90%) | 38.70 (-2.15%) |
| *w/o ensemble* | 28.81 (-7.00%) | 48.46 (-5.43%) | 37.00 (-21.38%) | 38.31 (-6.79%) | 34.76 (-10.44%) | 32.51 (-17.80%) |

### 4.2.5 ABLATION STUDY

We perform ablation experiments on six typical dataset to confirm the contribution of each operation in the GENOME+. As shown in Table 5, every operator of GENOME+ has a positive impact on performance. For comparison, we use *random selection* in place of selection because eliminating it completely would result in an unlimited growth of the population size. The replacement of our selection with random selection results in an average performance loss of 10.57%. For the ensemble operation, its removal results in an average performance drop of 11.47%, especially a steep reduction in 21.38% in the DROP. The removal of initialization and succession operations lead to performance drops of 5.87% and 5.97%, respectively. The crossover and mutation operations are crucial for maintaining population diversity, despite their modest consequences (performance decreases of 4.23% and 3.95%, respectively). When both operations are removed simultaneously, the performance drop enlarges to 12.47%, indicating a synergistic effect between them, eliminating both reduces population diversity and thus leads to a greater decline in overall performance.

### 4.2.6 TIME EFFICIENCY ON DIFFERENT GPUS

Table 6: Comparison of time and score on different GPUs

| Method | Task | Score | | Time (s) | |
|--------|------|-------|----|----------|----|
| | | **4090** | **A100** | **4090** | **A100** |
| **GENOME** | MMLUPro | **28.97** | 28.67 | ∼3600 | ∼1600 |
| | DROP | **35.80** | 35.32 | ∼1000 | ∼800 |
| | MATH | 15.56 | **15.78** | ∼4000 | ∼2600 |
| **GENOME+** | MMLUPro | **31.47** | 30.98 | ∼5000 | ∼3000 |
| | DROP | 46.20 | **46.92** | ∼2000 | ∼1200 |
| | MATH | 16.79 | **16.81** | ∼6000 | ∼3800 |

To evaluate the time efficiency of our method, we conduct experiments on two different GPUs: **RTX 4090 (24GB VRAM)** and **A100 (80GB VRAM)**. All experiments run on a single GPU and are repeated five times to obtain the average result. The base model is gemma-2-2b-it, and the inference framework is vLLM. The GPU memory utilization is set to 95% on both the A100 and 4090 to ensure a fair comparison. The population size set in 10, and other hyperparameters remain consistent with those used in the single-task experiments. As shown in Table 6, the inference speed of the 4090 and A100 varies across different tasks. Despite having less VRAM, the 4090 maintains high inference

---

[1]https://cloud.google.com/vertex-ai/generative-ai/docs/learn/models

performance on MMLUPro, DROP, and MATH tasks, further validating the applicability of our method. It is important to note that **CPU performance**, **memory bandwidth**, and **system load** may also impact inference speed. Therefore, while our experimental results provide an overall trend, variations may occur in different environments.

## 5 CONCLUSION

This paper formally defines the population-based evolution problem for LLMs and presents two novel solutions: GENOME and GENOME+. These two frameworks leverage principles of biological evolution to adapt LLMs to new tasks with the use of few samples. Our experiments indicate that GENOME+ consistently surpasses existing methods across various tasks, showing notable enhancements in reasoning-intensive tasks. The framework demonstrates strong multi-task domain capabilities and effective generalization to novel tasks, while ensuring scalability with larger population sizes (up to 40). Ablation studies demonstrate the essential role of each evolutionary component, with selection and ensemble operations exhibiting the greatest influence on performance. This study illustrates that population-based evolution presents a promising approach to enhance LLMs' capabilities through evolutionary optimization. In addition, our experiments confirm that GENOME(+) can operate on devices equipped with 24GB of VRAM. We open-source our code and release the expert models to enable reproducibility and encourage further research in this area.

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

# A APPENDIX

## A.1 ALGORITHM

The pseudocode for GENOME and GENOME(+) can be found in Algorithm 1 and 2.

---

**Algorithm 1 GENOME**

---

**Input:** Task $T$, fitness function $f$, LLM experts $\{x_i\}_{i=1}^n$
**Hyperparameters**: population size $N$, crossover rate $c_r$, mutation rates $(im_r, gm_r)$, sigma $\sigma$, elite ratio $\alpha$, max iteration $K$
*// Initialization*
$P \leftarrow \{\mathbf{w}_i = t\mathbf{w}_a + (1-t)\mathbf{w}_b | t \sim U(0,1)\}_{i=1}^N$
evaluate fitness $f(x_i)$ for $x_i \in P$
$g \leftarrow \arg\max_{x \in P} f(x)$
**for** $iter = 1$ **to** $K$ **do**
  *// Crossover*
  **for** each pair selected with probability $c_r$ **do**
    $p_i \leftarrow f_i / \sum f_k, \quad k = 1, 2, \cdots, N$
    select parents $(x_{p_1}, x_{p_2})$ according to $\{p_i\}_i^N$
    $\mathbf{w}_{child} \leftarrow t\mathbf{w}_{p_1} + (1-t)\mathbf{w}_{p_2}$, create $x_{child}$ by $\mathbf{w}_{child}$
    evaluate $x_{child}$ on $f$
  **end for**
  *// Mutation*
  **for** each individual with probability $im_r$ **do**
    **for** each weight $\omega_{ij}$ with probability $gm_r$ **do**
      $\mathbf{w}_i' \leftarrow \mathbf{w}_i + \mathbf{m}_i \cdot \mathbf{E}_i$, create $x_i'$ by $\mathbf{w}_i$
      evaluate $x_i'$ on $f$
    **end for**
  **end for**
  *// Selection*
  $P \leftarrow \text{elite}(\alpha \cdot N) \cup \text{fitness-based selection}((1-\alpha) \cdot N)$
  $g \leftarrow \arg\max_{x \in P} f(x)$
**end for**
evaluate test fitness $f_{\text{test}}(\mathbf{x}_i)$ for $\mathbf{x}_i \in P$, update $\mathbf{g}$
**return** best individual $g$

---

## A.2 DATASETS AND EXPERT MODELS

**Datasets** To ensure robust evaluation, we randomly sample 200 instances from each dataset as the validation set for model optimization. For the remaining data, we employ a size-based splitting strategy: for datasets with fewer than 1,000 remaining instances, we use all data as the test set; for those exceeding 1,000 instances, we retain 1,000 instances as the test set, except for multilingual tasks. For multilingual tasks (MGSM, FLORES-101), considering the scarcity of low-resource languages, we retain all remaining data as the test set to avoid potential language bias. Additionally, for FLORES-101, we create a new **FLORES-37** dataset by selecting data corresponding to the 37

---

**Algorithm 2 GENOME+**

---

**Input:** Task $T$, fitness function $f$, LLM experts $\{\mathbf{x}_i\}_{i=1}^n$
**Hyperparameters:** population size $N$, crossover rate $c_r$, mutation rates $(im_r, gm_r)$, sigma $\sigma$, elite ratio $\alpha$, learning weight $\phi_*$, learning rate $\lambda$, max iteration $K$
**Population Initialization**
**for** $iter = 1$ **to** $K$ **do**
   **Crossover, Mutation**
   *// succession*
   **for** $i = 1$ **to** $N$ **do**
      $\mathbf{e}_i \leftarrow \frac{1}{\mathbf{C}} [\phi_e \mathbf{e}_i + \phi_g(\mathbf{e}_g - \mathbf{e}_i) + \phi_c(\mathbf{e}_c - \mathbf{e}_i) - \phi_w(\mathbf{e}_w - \mathbf{e}_i)]$
      $\mathbf{w}_i \leftarrow \mathbf{w}_i + \lambda e_i$, create $x_i$ by $\mathbf{w}_i$
   **end for**
   **Selection**
**end for**
evaluate test fitness $f_{\text{test}}$ for $x_i \in P$, update $\mathbf{g}$
*// ensemble*
Select top-$k$ individuals based on validation fitness $f_{\text{valid}}$
Combine selected individuals into an ensemble $\mathbf{E}$
**return** best individual $\mathbf{g}$ and ensemble $\mathbf{E}$

---

languages supported by the *gemma-2* model (Team et al., 2024). Furthermore, compared to the best single experiment, in the multi-task experiment, we utilize the MELD dataset, which, together with EmoryNLP, forms the tasks in the Affective Computing.

Table 7: Detailed information of the datasets.

| Dataset | Category | Metrics | Size | |
|---------|----------|---------|------------|------|
| | | | validation | test |
| ARC_C | Question Answering | accuracy, 0-shot | 200 | 1000 |
| BBH | Logical Reasoning | accuracy, 3-shot | 200 | 1000 |
| CSQA | Question Answering | accuracy, 0-shot | 200 | 1000 |
| DROP | Logical Reasoning | exact match, 0-shot | 200 | 1000 |
| EmoryNLP | Affective Computing | accuracy, 0-shot | 200 | 697 |
| Flores-37/101 | Multilingual Processing | BLEU, 3-shot | 200 | 1012 |
| GSM8k | Mathematics | accuracy, 0-shot | 200 | 1000 |
| MATH | Mathematics | accuracy, 0-shot | 200 | 1000 |
| MBPP | Code Generation | Pass@1, 0-shot | 200 | 774 |
| MELD | Affective Computing | accuracy, 0-shot | 200 | 1000 |
| MGSM | Multilingual Processing, Mathematics | accuracy, 0-shot | 200 | 2637 |
| MMLU | General Knowledge | accuracy, 0-shot | 200 | 1000 |
| MMLUPro | General Knowledge | accuracy, 0-shot | 200 | 1000 |

**Expert Models** All models are trained on $8\times$A100-80GB GPUs. We visualize the performances (ranks) of our ten initial expert models across all datasets, grouped by category, in Figure 3. It is evident that each model exhibits distinct strengths, demonstrating that we have successfully trained a diverse set of experts.

### A.3 IMPLEMENTATION DETAILS

**Ensemble** For tasks with a single correct answer, we use majority voting to aggregate predictions. For tasks without a unique answer (e.g., MBPP, Flores37/101, DROP), we apply a similarity-based approach. Specifically, we take the top-k outputs, embed each using `text-embedding-3-small`, and compute pairwise similarities via BERTScore (Zhang et al., 2019). The output with the highest overall similarity score is selected as the final ensemble result.

**Baseline** To ensure fair comparisons, we use the same set of 10 initial expert models for all baselines except for the data merge baseline, and we otherwise align our settings as closely as possible

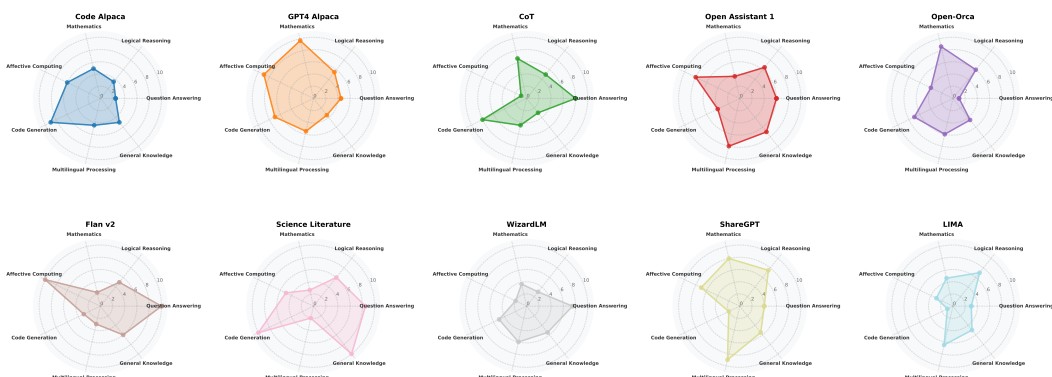

Figure 3: Capability distribution of expert models across seven dimensions, highlighting their specialized strengths.

Table 8: Training configurations and sample sizes for different subsets. The learning rate decay follows a cosine schedule.

| Subset | Samples | LoRA rank/alpha | Learning rate | Warmup ratio | Batchsize | Epochs |
|---|---|---|---|---|---|---|
| CoT (Chung et al., 2024) | 49747 | 8/16 | 2.00E-04 | 0.1 | 32 | 5 |
| Code Alpaca (Chaudhary, 2023) | 20016 | 8/16 | 2.00E-04 | 0.1 | 32 | 5 |
| Flan v2 (Chung et al., 2024) | 49123 | 8/16 | 2.00E-04 | 0.1 | 32 | 5 |
| GPT4 Alpaca (Peng et al., 2023) | 19906 | 8/16 | 2.00E-04 | 0.1 | 32 | 5 |
| Open Assistant 1 (Köpf et al., 2024) | 7331 | 8/16 | 2.00E-04 | 0.1 | 32 | 5 |
| Open-Orca (Mukherjee et al., 2023) | 29683 | 8/16 | 2.00E-04 | 0.1 | 32 | 5 |
| Science Literature (Ivison et al., 2023a) | 7468 | 8/16 | 2.00E-04 | 0.1 | 32 | 5 |
| ShareGPT (TechCrunch, 2023) | 111912 | 8/16 | 2.00E-04 | 0.1 | 32 | 1 |
| WizardLM (Xu et al., 2023) | 29810 | 8/16 | 2.00E-04 | 0.1 | 32 | 5 |
| LIMA (Zhou et al., 2023) | 1018 | 8/16 | 2.00E-04 | 0.1 | 32 | 5 |
| Tulu-v2-sft-mixture (Ivison et al., 2023b) | 326014 | 8/16 | 2.00E-04 | 0.1 | 32 | 1 |

with the original papers. LoRAHub is run for 50 iterations, and Model Swarms uses a population size of 10 for 10 iterations, with other hyperparameters set according to its original recommendations. Unless otherwise specified, all experiments are performed on 4×A100 80G GPUs and repeated 5 times.

**Prompts**  All methods in our experiments rely on the same prompts for a fair comparison. Following OpenAI's approach[2], we design specific prompts for different tasks while maintaining a minimalist design philosophy. All prompts are shown in Figure 4-14. As MMLUPro questions can have more than four options, we implement a more flexible prompt structure (Figure 5) to accommodate the varying number of choices.

## A.4 HYPERPARAMETER STATISTICAL ANALYSIS

To optimize hyperparameter selection and verify the stability of the GENOME and GENOME+ methods, we conduct 500 experiments. The hyperparameters involved in the study include:

- Cross rate ($c_r$): Range from 0.1 to 1.0.
- Individual mutation rate ($im_r$): Range from 0.1 to 1.0.
- Gene mutation rate ($gm_r$): Range from 0.1 to 1.0.

The dependent variables are *test performance*, *Top-3 ensemble performance*, and *optimization time*. The experiments use a fixed random seed of 47, set the population size (N) to 10, $\sigma$ to 0.001, and selected GENOME as the test method (we believe that the hyperparameter settings of GENOME and GENOME+ are similar). The dataset use in this experiment is MMLUpro, which is a highly comprehensive test dataset. The experimental environment consisted of $4 \times A100$ 80G GPUs. In

---

[2]https://github.com/openai/simple-evals

---

**MMLU**

```
Answer the following {subject} question.
The last line of your response should be
of the following format:
'Answer: $LETTER' (without quotes)
where LETTER is one of ABCD.

{question}

A) {A}
B) {B}
C) {C}
D) {D}

Let's think step by step.
```

Figure 4: The prompt of MMLU.

---

**MMLUPro**

```
Answer the following {subject} question.
The last line of your response should be
of the following format:
'Answer: $LETTER' (without quotes)
where LETTER is one of {candidates}.

{question}

{options}

Let's think step by step.
```

Figure 5: The prompt of MMLUPro.

---

**ARC_C**

```
Answer the following question.
The last line of your response should be
of the following format:
'Answer: $LETTER' (without quotes)
where LETTER is one of {candidates}.

{question}

{options}

Let's think step by step.
```

Figure 6: The prompt of ARC_C.

---

**CSQA**

```
Answer the following {question_type}
question step by step.

The last line of your response should be
of the following format:
'Answer: $LETTER' (without quotes)
where LETTER is one of ABCDE.

{question}

{options}
```

Figure 7: The prompt of CSQA.

---

**BBH**

```
Here are some examples to help you
understand how to answer the question.

{examples}

Now answer the following question,
write a line of the form "Answer: $Answer"
at the end of your response.

{question}
```

Figure 8: The prompt of BBH.

---

**Flores-37/101**

```
Translate the following sentence from English
to {language}.

Your output should be formatted as follows:

Translation: $SENTENCE

(where $SENTENCE is the translated version
of the sentence into {language}).

Below are examples to guide the translation
task:

{examples}

Now translate the following sentence:

{sentence}
```

Figure 9: The prompt of Flores-37/101.

---

**GSM8K / MATH**

```
Solve the following math problem step by step.

The last line of your response should be the form
Answer: $ANSWER (without quotes) where $ANSWER is
the answer to the problem.

{Question}

Remember to put your answer on its own line after
"Answer:", and you do not need to use a \\boxed
command.
```

Figure 10: The prompt of GSM8K and MATH.

**MGSM**

```
Solve the following math problem step by step.

Provide your explanation and final answer in English,
the last line of your response should be the form
Answer: $ANSWER (without quotes) where $ANSWER is
the answer to the problem.

{Question}

Remember to put your answer on its own line after
"Answer:", and you do not need to use a \\boxed
command, always respond in English.
```

Figure 11: The prompt of MGSM.

**MBPP**

```
You are an expert Python programmer,
and here is your task:

{question}

Your code should pass these tests:

{test}
```

Figure 12: The prompt of MBPP.

**DROP**

```
You will be asked to read a passage and
answer a question.

Write a line of the form "Answer: $ANSWER"
at the end of your response.

{context}
```

Figure 13: The prompt of DROP.

**EmoryNLP / MELD**

```
Given a conversation history and a current utterance,
follow these steps to identify the emotion of the
current utterance from the given options.

The emotion should be determined based on both the
conversation context and the current utterance.

The last line of your response should be of the
following format: 'Answer: $LETTER' (without quotes)
where LETTER is one of ABCDEFG.

Let's think step by step.

History:
{history}

Utterance:
{utterance}

Options:
{options}
```

Figure 14: The prompt of EmoryNLP and MELD.

each experiment, we discretize each hyperparameter range into fixed intervals of 0.1 (e.g., 0.1, 0.2, 0.3, ..., 1.0). From these discrete candidate values, one value was randomly selected for $c_r$, $im_r$, and $gm_r$ in each experimental run, subject to the same random seed initialization.

In the correlation analysis (see in Table 9), Spearman and Pearson methods reveal that the $c_r$ had a significant positive correlation with *test performance* and *optimization time*, and a weak correlation with *ensemble performance*. The $im_r$ is found to be weakly negatively correlated with *test performance* but significantly positively correlated with *optimization time*. The $gm_r$ show small correlation coefficients with all dependent variables.

Table 9: Correlation Analysis between Algorithm Parameters and Performance Metrics

| HyperParameter | Metric | Pearson | | Spearman | |
|---|---|---|---|---|---|
| | | $r$ | p-value | $\rho$ | p-value |
| Cross Rate | Test | 0.192*** | 2.132e-03 | 0.194*** | 1.927e-03 |
| | Ensemble | 0.119* | 5.831e-02 | 0.102 | 1.052e-01 |
| | Time | 0.382*** | 3.030e-10 | 0.408*** | 1.395e-11 |
| Individual Mutation Rate | Test | -0.134** | 3.284e-02 | -0.118* | 5.946e-02 |
| | Ensemble | -0.091 | 1.479e-01 | -0.094 | 1.368e-01 |
| | Time | 0.775*** | 4.647e-52 | 0.796*** | 9.183e-57 |
| Gene Mutation Rate | Test | -0.093 | 1.395e-01 | -0.092 | 1.433e-01 |
| | Ensemble | 0.077 | 2.240e-01 | 0.092 | 1.423e-01 |
| | Time | 0.123* | 5.002e-02 | 0.125** | 4.684e-02 |

*Notes: *** $p < 0.01$, ** $p < 0.05$, * $p < 0.1$*

Given the complex coupling relationships between variables (e.g., gene mutation rate is only meaningful when individual mutation rate is greater than 0), simple correlation analysis may not be sufficient. Therefore, we conduct further multiple linear regression analysis. For example, for *test performance*, we construct the following regression model:

$$\text{test}_i = \beta_0 + \beta_1 \cdot c_{ri} + \beta_2 \cdot im_{ri} + \beta_3 \cdot gm_{ri} + \varepsilon_i \tag{6}$$

We construct corresponding regression models for *ensemble performance* and *optimization time* and estimate the parameters using the least squares method. The significance test results are shown in Table 10.

Analysis results show that $c_r$ has a significant positive impact on *test performance*, and $im_r$ shows a negative impact. The impact of $gm_r$ do not reach significance. For *ensemble performance*, both correlation and multiple regression analyses show that the effects of the three hyperparameters are not significant, indicating that after adopting the ensemble strategy, the GENOME's sensitivity to hyperparameters decreased. In terms of *optimization time*, increased cross rate and individual mutation rate significantly prolonged total runtime.

Combining the results of correlation and regression analysis, it can be concluded that the $c_r$ plays a significant role in improving the quality of solutions (test performance) generated by GENOME, though it also increases computational runtime. This finding suggests that a higher $c_r$ enhances the combination quality of chromosomes (model weights), leading to better solutions, albeit at the cost of greater computational effort. On the other hand, an excessively high $im_r$ tends to disrupt superior individuals and substantially enlarge the search space, leading to increased time consumption. However, a moderate level of $im_r$ proves beneficial as it helps prevent the population from prematurely converging. In contrast, the $gm_r$ demonstrate only a limited impact within the scope of this study, implying its influence on the GENOME's performance is relatively minor.

Within a broad range of hyperparameter settings, the performance of GENOME is relatively stable, suggesting it is not sensitive to hyperparameter values. The introduction of the ensemble strategy further reduces sensitivity, effectively balancing different parameter choices while maintaining solution quality.

The recommended hyperparameter settings are a $c_r$ of 0.3 to 0.6, $im_r$ of 0.1, and $gm_r$ of 0.1. This setup can build superior individuals while maintaining population diversity and avoiding the negative effects of excessive mutation. Even in larger-scale or more complex problem scenarios, this

Table 10: Regression Results for Genetic Algorithm Parameters

| Variable | Coefficient | Std. Error | t-statistic | p-value |
|---|---|---|---|---|
| **Panel A: Test Performance** | | | | |
| Constant | 0.2881*** | 0.001 | 198.057 | 0.000 |
| Cross Rate | 0.0054*** | 0.002 | 3.279 | 0.001 |
| Individual Mutation Rate | -0.0030** | 0.002 | -1.974 | 0.049 |
| Gene Mutation Rate | -0.0028* | 0.002 | -1.826 | 0.069 |
| **Panel B: Ensemble Performance** | | | | |
| Constant | 0.2985*** | 0.002 | 150.732 | 0.000 |
| Cross Rate | 0.0038* | 0.002 | 1.700 | 0.090 |
| Individual Mutation Rate | -0.0030 | 0.002 | -1.455 | 0.147 |
| Gene Mutation Rate | 0.002 | 0.002 | 1.071 | 0.285 |
| **Panel C: Computational Time (s)** | | | | |
| Constant | 507.28*** | 116.53 | 4.353 | 0.000 |
| Cross Rate | 1724.37*** | 130.94 | 13.169 | 0.000 |
| Individual Mutation Rate | 3159.24*** | 122.64 | 25.760 | 0.000 |
| Gene Mutation Rate | 111.91 | 122.96 | 0.910 | 0.364 |

*Notes:* *** $p < 0.01$, ** $p < 0.05$, * $p < 0.1$

Table 11: Performance comparison of different methods on MMLUPro knowledge and reasoning subsets. The percentages in parentheses indicate the relative improvement (red) or degradation (green) compared to the Best Single.

| Method | MMLUPro$_{Knowledge}$ | MMLUPro$_{Reasoning}$ |
|---|---|---|
| **Best Single** | 27.90 | 23.20 |
| **Model Swarms** | 27.36 (-1.94%) | 24.95 (+7.54%) |
| **GENOME** | 27.48 (+0.44%) | 25.24 (+8.79%) |
| **GENOME+** | 29.05 (+6.18%) | 28.73 (+23.84%) |

configuration, through the ensemble strategy, can smooth out parameter differences and fully leverage the core advantages of GENOME, achieving relatively stable overall performance.

## A.5 FURTHER EXPERIMENTS

**Variance Across Random Seeds** For completeness, we report the variability of all main methods across random seeds. All numbers in the main text are means over five runs with different random seeds $\{41, 42, 47, 53, 3407\}$. To keep the main tables readable (12 datasets and multiple methods), we only show mean scores there, which is also consistent with prior work we compare to (e.g., LoRAHub, Pack of LLMs, MERGE[3], and Model Swarms). Table 12 provides the full "mean $\pm$ standard deviation" statistics for all methods and datasets.

**Robustness to Validation Set Size** We examine how sensitive our evolution procedure is to the choice and size of the validation set. Following standard practice in adapter-based model-merging work, we use a relatively small held-out set (200 examples per task) to guide search/merging, and a larger, disjoint test set for final reporting. In our setup, for each task: (i) the validation set is randomly sampled from the benchmark and used only for evolution/selection; and (ii) the test set is strictly disjoint and all main-table results are computed on this test set. To directly address concerns about potential overfitting to a particular 200-example split, we vary the validation size $N \in \{20, 50, 100, 150, 200\}$ and resample the validation set while keeping the test set fixed. For three representative tasks (MMLUPro, MATH, DROP), we run GENOME and GENOME+ and report mean $\pm$ standard deviation over 5 seeds (Table 13–15). The $N = 200$ columns are directly copied from the main experiments. We observe that (i) very small validation sets (e.g., $N = 20$) lead to somewhat less stable performance, which is expected; and (ii) as $N$ increases, both GENOME and GENOME+ show only mild variation, with GENOME+ consistently outperforming GENOME

Table 12: Per-dataset mean ± standard deviation over five random seeds for all main methods.

| Method | MMLU | MMLUPro | GSM8K | MATH | MGSM | Flores-37 | ARC_C | CSQA | BBH | DROP | EmoryNLP | MBPP |
|--------|------|---------|-------|------|------|-----------|-------|------|-----|------|----------|------|
| LoRAHub | 53.00±0.78 | 27.17±1.57 | 46.47±3.35 | 14.90±0.89 | 34.70±2.80 | 21.83±1.12 | 69.97±3.58 | 66.10±1.01 | 39.30±3.13 | 35.32±5.59 | 31.53±0.89 | 42.63±1.79 |
| MERGE$^3$-LoRA | 53.16±0.70 | 28.09±2.08 | 48.03±0.84 | 14.90±1.57 | 37.81±5.20 | 22.19±2.00 | 70.01±1.37 | 67.68±4.53 | 39.40±2.40 | 33.28±4.68 | 32.77±2.09 | 42.48±1.03 |
| Pack of LLMs | 54.30±2.46 | 27.67±0.78 | 38.13±1.01 | 11.50±2.91 | 31.06±1.70 | 19.24±2.35 | 70.22±1.12 | 63.30±3.13 | 38.90±0.89 | 21.30±4.25 | 30.46±2.80 | 42.56±1.01 |
| Model Swarms | 55.91±1.07 | 27.77±1.69 | 45.82±2.49 | 15.06±2.38 | 33.15±2.70 | 21.76±2.57 | 68.53±2.94 | 68.76±2.09 | 38.80±3.23 | 34.52±5.90 | 33.98±2.91 | 42.56±2.65 |
| GENOME | 56.66±0.93 | 27.77±2.20 | 49.34±2.04 | 15.72±1.66 | 37.19±1.79 | 22.78±1.91 | 70.69±2.50 | 68.22±2.71 | 40.00±2.10 | 35.28±6.31 | 34.97±2.51 | 43.60±1.86 |
| GENOME+ | 56.44±0.63 | 30.98±1.01 | 51.24±1.34 | 16.41±1.32 | 39.55±1.57 | 23.38±1.24 | 74.38±1.45 | 69.89±1.12 | 41.10±1.57 | 47.06±5.37 | 38.81±1.45 | 43.54±1.34 |

Table 13: Validation-set size study on MMLUPro: mean ± std over 5 seeds.

| Method | $N$=20 | $N$=50 | $N$=100 | $N$=150 | $N$=200 |
|--------|--------|--------|---------|---------|---------|
| GENOME | 26.27±1.72 | 26.57±2.52 | 27.50±1.74 | 27.97±1.75 | 27.77±2.20 |
| GENOME+ | 29.97±1.26 | 30.09±1.10 | 30.42±0.69 | 30.74±1.02 | 30.98±1.01 |

across all $N$. Overall, we see no evidence that our results depend on a particular 200-example valida-tion split: the final test-set performance is robust to both validation set size and random resampling.

**Other Base Model**  We used Llama-3.1-8B-Instruct (Grattafiori et al., 2024) and Qwen2.5-14B-Instruct (Qwen et al., 2025) as the base model (fine-tuned with the same SFT dataset for LoRA weights) on three representative tasks (MMLUPro, MATH, DROP) to compare GENOME(+). The results are shown in Table 16 and 17 . GENOME(+) still achieved the best performance. This indicates that GENOME(+) is not only effective for 2B models, but also works well on 8B model and 14B model.

**Subtask**  To verify whether this difference is influenced by task type, we construct subsets of knowledge-based and reasoning-based tasks within the MMLUPro. The former includes tasks that rely on domain knowledge such as law, history, and psychology, while the latter comprises fields that require logical reasoning, such as mathematics, physics, and computer science. As shown in Table 11, the improvement rates of all methods on reasoning-based tasks are much higher than those on knowledge-based tasks. Notably, GENOME+ achieves a performance gain of 23.84% on the reasoning subset compared to the Best Single, whereas the gain is 6.18% on the knowledge subset. Other methods exhibit a similar trend. These results reveal that existing dynamic methods, including our population-based evolution framework, excels in reasoning-intensive tasks. On the other hand, knowledge-intensive tasks may require the integration of external knowledge bases or knowledge enhancement strategies, which is beyond the capability of dynamic methods.

**Additional Baselines: Fine-tuning and Direct Ensembles**  To complement the main baselines, we further evaluate several additional settings for instruction tuning. The benchmarks we use are standard LLM evaluation suites that only provide inputs and final answers, and generally lack inter-mediate reasoning traces or consistent output formats. Direct LoRA fine-tuning on such raw (input, answer) pairs is known to be weak supervision for reasoning-heavy tasks and also leads to format-parsing issues at evaluation time. We therefore construct two fine-tuning datasets and train LoRA baselines on Gemma-2-2B-it under the same hyperparameters as our original experts:

• **Raw benchmark fine-tuning**: LoRA fine-tuning directly on the original (question, answer) benchmark pairs (with 200 validation examples per task).

• **Distilled benchmark fine-tuning**: a strong teacher model (GPT-5) is used to (i) generate inter-mediate reasoning and (ii) normalize answer formats; Gemma-2-2B-it is then fine-tuned on this enhanced dataset. Note that this baseline benefits from additional teacher supervision, while our method and all model-merging baselines do not use any external model supervision.

For the requested ensemble-style baselines:

• **Direct Top-3 Ensemble**: a top-3 ensemble of the original experts, matching the main text setup (same $k$ and prompts), without any evolutionary search.

• **Expert Fusion (naive averaging)**: this corresponds exactly to the expert-fusion baseline reported in Tables 1–4 (naive parameter averaging of top experts, without evolution).

Table 14: Validation-set size study on MATH: mean ± std over 5 seeds.

| Method | $N$=20 | $N$=50 | $N$=100 | $N$=150 | $N$=200 |
|---|---|---|---|---|---|
| GENOME | 15.83±1.32 | 16.13±0.90 | 15.63±1.45 | 15.07±1.07 | 15.72±1.66 |
| GENOME+ | 15.38±0.88 | 15.83±1.70 | 16.48±1.35 | 17.67±0.73 | 16.41±1.32 |

Table 15: Validation-set size study on DROP: mean ± std over 5 seeds.

| Method | $N$=20 | $N$=50 | $N$=100 | $N$=150 | $N$=200 |
|---|---|---|---|---|---|
| GENOME | 33.75±6.75 | 34.27±2.41 | 35.03±7.23 | 33.70±3.76 | 35.28±6.31 |
| GENOME+ | 45.73±5.75 | 47.97±5.28 | 45.03±5.73 | 45.40±5.15 | 47.06±5.37 |

Table 16: The performance of single task (base model: Llama-3.1-8B-Instruct).

| Method | MMLUPro | MATH | DROP |
|---|---|---|---|
| **Best Single** | 45.00 | 24.00 | 50.00 |
| **Data Merge** | 28.47 | 24.80 | 33.99 |
| **DARE_TIES** | 46.45 | 30.00 | 32.30 |
| **Expert Fusion** | 45.85 | 31.60 | 34.10 |
| **LoraHub** | 43.46 | 31.60 | 49.60 |
| **Pack of LLMs** | 45.45 | 31.40 | 21.90 |
| **Model Swarms** | 45.50 | 32.20 | 53.70 |
| **GENOME** | 48.15 | 32.65 | 56.20 |
| *- vs Best Single* | +7.00% | +36.04% | +12.40% |
| *- vs Model Swarms* | +5.82% | +1.40% | +4.64% |
| **GENOME+** | **49.15** | **34.55** | **60.10** |
| *- vs Best Single* | +9.22% | +43.96% | +20.20% |
| *- vs Model Swarms* | +8.02% | +7.30% | +11.92% |

Table 17: The performance of single task (base model: Qwen2.5-14B-Instruct).

| Method | MMLUPro | MATH | DROP | MGSM |
|---|---|---|---|---|
| **Best Single** | 58.74 | 36.60 | 61.20 | 65.49 |
| **Data Merge** | 53.50 | 35.50 | 19.80 | 59.54 |
| **Expert Fusion** | 59.14 | 34.30 | 54.30 | 53.20 |
| **LoraHub** | 58.27 | 39.20 | 66.27 | 64.95 |
| **Pack of LLMs** | 58.44 | 28.60 | 42.33 | 51.02 |
| **Model Swarms** | 60.24 | 42.00 | 64.87 | 66.64 |
| **GENOME** | 61.02 | 42.23 | 66.60 | 66.77 |
| *- vs Best Single* | +3.88% | +15.38% | +8.82% | +1.95% |
| *- vs Model Swarms* | +1.29% | +0.55% | +2.67% | +0.20% |
| **GENOME+** | **64.09** | **43.20** | **79.20** | **67.48** |
| *- vs Best Single* | +9.11% | +18.03% | +29.41% | +3.04% |
| *- vs Model Swarms* | +6.39% | +2.86% | +22.09% | +1.26% |

Since these additional baselines do not involve stochastic evolutionary operations, we report them under a single seed (seed = 42). For comparison, we also include the 5-seed averages of GENOME and GENOME+. Results are summarized in Table 18. We observe three main trends. (i) *Raw fine-tuning* performs the worst overall and even collapses to 0.00 on several datasets, consistent with the intuition that using only (input, final answer) pairs without reasoning traces and with noisy formats provides a very weak supervision signal. (ii) Even with *GPT-5 distillation*, gains over GENOME+ are limited: the distilled fine-tuning baseline benefits from an extra powerful teacher, yet it surpasses GENOME+ on only a few datasets, while GENOME+ never relies on any external model. (iii) *Direct aggregation* (Top-3 Ensemble and naive Expert Fusion) is highly task-dependent: it helps on some

Table 18: Additional baselines on Gemma-2-2B-it. All numbers are single-seed results (seed = 42) except for GENOME / GENOME+, which are averaged over 5 seeds.

| Setting | MMLU | MMLUPro | GSM8K | MATH | MGSM | Flores-37 | ARC_C | CSQA | BBH | DROP | EmoryNLP | MBPP |
|---|---|---|---|---|---|---|---|---|---|---|---|---|
| Raw fine-tuning | 26.92 | 10.00 | 0.00 | 0.00 | 0.00 | 0.00 | 26.09 | 20.00 | 0.00 | 47.50 | 1.33 | 39.50 |
| Distilled fine-tuning | **56.73** | 28.50 | 46.50 | 12.00 | 32.50 | 22.75 | **74.58** | 68.50 | **43.00** | **54.50** | 36.72 | 35.00 |
| Direct Top-3 Ensemble | 48.80 | 25.82 | 42.45 | 12.55 | 27.68 | 19.36 | 68.98 | 67.32 | 32.43 | 19.75 | 36.89 | 31.78 |
| Expert Fusion | 55.80 | 27.67 | 39.50 | 13.10 | 31.82 | 21.74 | 69.03 | 65.00 | 31.82 | 22.60 | 32.66 | 28.55 |
| GENOME | 56.66 | 27.77 | 49.34 | 15.72 | 37.19 | 22.78 | 70.69 | 68.22 | 40.00 | 35.28 | 34.97 | **43.60** |
| GENOME+ | 56.44 | **30.98** | **51.24** | **16.41** | **39.55** | **23.38** | 74.38 | **69.89** | 41.10 | 47.06 | **38.81** | 43.54 |

benchmarks but degrades sharply on others, illustrating that naive expert aggregation is unstable compared with the evolutionary GENOME/GENOME+ approach.

## A.6    LIMITATION

Our proposed method, GENOME(+), is gradient-free and can run on an NVIDIA RTX 4090. However, its optimisation requires multiple LoRA adapters that must be pre-trained. In addition, like most model-merging techniques, our approach currently supports only the fusion of homogeneous models.

## A.7    REPRODUCIBILITY STATEMENT

In the anonymous supplementary materials, we provide the complete source code and run scripts; we also release our source code on github: `https://anonymous.4open.science/r/GENOME-0DBD`; the implementation and pseudocode correspond to §3 (Algorithm 1 and Algorithm 2, Figure 1). The experimental setup (base models, expert construction pipeline, and training configurations) is detailed in §4.1 and Table 9; all default/searched hyperparameters and their ranges are given in §4.1 and Appendix §A.3. Datasets, splits, and evaluation metrics are specified in §4.1 and Table 7 (for each task, we fix 200 examples drawn from the original data as a validation set for evolution/selection, and report results on test sets of about 1,000 examples; criteria and templates for multilingual and tasks without a unique answer are in Figure 4–14). We use five different random seeds across all experiments and report the mean (§4). Evaluation uses a unified prompt template (Figure 4–14); details of the ensembling strategy and similarity assessment are provided in §A.3. Regarding compute, main results are reproduced on 4×A100 80 GB GPUs, and key experiments can run on a single RTX 4090 (24 GB).

