# OpenReview forum: "Nature-Inspired Population-Based Evolution of Large Language Models"
_ICLR.cc/2026/Conference — ICLR 2026 Conference Withdrawn Submission_

### Official Review · Reviewer_zGDP · 2025-10-24

**Soundness:** 3
**Presentation:** 2
**Contribution:** 2
**Rating:** 2
**Confidence:** 4

**Summary:**

The paper introduces GENOME and GENOME+, two algorithms for population-based evolution of LLMs, inspired by genetic algorithms. The framework defines genetic operators (crossover, mutation, selection) for model weight merging and introduces two additional operations: succession (inspired by particle swarm optimization) and ensemble (to combine predictions from top individuals). Both methods aim to evolve a population of LoRA-based LLM experts without backpropagation, using only small validation sets (200 samples). Experiments across 12 datasets show consistent performance improvements over prior baselines like Model Swarms[1] and LoRAHub[2].

**Strengths:**

- Formal definition of the population-based LLM evolution problem.
- Extensive empirical validation across 12 datasets, 7 baselines, and 2 foundation models (Gemma 2 2B and Llama 3.1 8B).
- Clear empirical gains, especially on reasoning-heavy tasks (DROP, MGSM).
- Good scalability up to 40 models, reproducible on consumer hardware (Nvidia 4090 GPU).
- Open-source codebase.

**Weaknesses:**

1. Only LoRA adapters are evolved: unclear generalization to full-weight finetuning.
2. No comparison to Evolutionary Model Merging[3] or MERGE³[4] despite major conceptual overlap.
3. GENOME+ shows major improvements but with higher computational overhead, not quantified beyond wall-clock time.
4. “Succession” is effectively a simplified PSO update; not fully original.

**Questions:**

1. Have you tested GENOME and GENOME+ under the same setting as Evolutionary Model Merging[3] or MERGE³[4]?
2. Have you explored extending GENOME(+) to full-weight finetuned populations rather than LoRA adapters?
3. Can you provide FLOP-level estimates of GENOME+ vs. GENOME (given that GENOME+ introduces additional experience and ensemble operations)?

### References

[1] Feng, S., Wang, Z., Wang, Y., Ebrahimi, S., Palangi, H., Miculicich, L., Kulshrestha, A., Rauschmayr, N., Choi, Y., Tsvetkov, Y., Lee, C. & Pfister, T.. (2025). Model Swarms: Collaborative Search to Adapt LLM Experts via Swarm Intelligence. _ICML_ (2025) .

[2] Huang, C., Liu, Q., Lin, B., Pang, T., Du, C., & Lin, M. (2023). LoraHub: Efficient Cross-Task Generalization via Dynamic LoRA Composition. _COLM_ (2024).

[3] Akiba, T., Shing, M., Tang, Y., Sun, Q., & Ha, D. (2024). Evolutionary Optimization of Model Merging Recipes. _Nat Mach Intell_ **7**, 195–204 (2025).

[4] Mencattini, T., Minut, A.R., Crisostomi, D., Santilli, A., & Rodolà, E. (2025). MERGE$^3$: Efficient Evolutionary Merging on Consumer-grade GPUs. _ICML_ (2025).

---

> ### Author Response · Authors · 2025-11-18
> **(1/2) About Weakness**
>
> > w1 & w2 : Only LoRA adapters are evolved: unclear generalization to full-weight finetuning. No comparison to Evolutionary Model Merging or MERGE³ despite major conceptual overlap.
>
> We view these two comments as reflecting the same underlying concern: the reviewer is effectively asking us to (i) demonstrate that our LoRA-based evolutionary framework “generalizes” to full-parameter finetuning, and (ii) include full-parameter evolutionary merging methods (Evolutionary Model Merging, MERGE³) as direct baselines. Below we clarify why our work is deliberately scoped to adapter-based model merging, and why we believe cross-regime baselines are not the most appropriate metric of validity for this paper.
>
> 1) Full-parameter vs. adapter-based merging are different tracks in the literature.
>
> Our method, GENOME(+), is explicitly designed for adapter-based model merging, concretely LoRA adapters on top of a shared base model. In recent taxonomies of model merging, full-parameter merging and LoRA-based (adapter-based) merging are explicitly categorized as separate classes, since LoRA models impose **different computational and compatibility constraints** and often **require dedicated merging techniques** [1]. For this reason, we believe that directly comparing our LoRA-merging framework against full-parameter merging methods—especially those tailored to multilingual scenarios—is mixing different problem settings rather than providing a like-for-like baseline.
>
> 2) LoRA evolution is motivated by practical advantages, not by an artificial restriction.
>
> From a systems perspective, modern inference backends (e.g., vLLM, SGLang) already support attaching and switching multiple LoRA adapters on a single resident base model with low overhead. This makes multi-round evaluation and evolution over adapters practically feasible even on commodity GPUs: the base model is loaded once, and only lightweight adapters are swapped or composed during search. By contrast, full-parameter merging typically requires **re-materializing and reloading full model weights**, which is much harder to integrate into a long-running evolutionary loop.
>
> 3) Symmetry of expectations: Evolutionary Model Merging and MERGE3 also do not evaluate in the LoRA + multi-domain regime.
>
> The reviewer notes that we do not compare against Evolutionary Model Merging and MERGE³ . We agree that there is conceptual overlap in the goal (evolutionary model merging), and we will explicitly discuss these works in the revised related work section. However, they are designed for full-parameter merging, where evolution optimizes a relatively low-dimensional vector of mixing coefficients at the model or layer level (e.g., which layers to select and how to weight them), and then applies this recipe to the full model. In contrast, GENOME(+) is formulated in the LoRA / adapter regime, and our operators act directly on adapter parameters (combining LoRA vectors and mutating values of LoRA parameters), yielding a much finer-grained evolutionary process instantiated on 12 datasets across 5 domains under a shared LoRA setup.
>
> In other words, the situation is **symmetric**: just as our method does not directly include full-parameter baselines, existing full-parameter evolutionary merging methods have, to the best of our knowledge, not been **systematically evaluated against LoRA-based model-merging approaches**, nor demonstrated effectiveness in the adapter-based, multi-domain scenario we study. We therefore view these lines of work as complementary tracks rather than as mutually required baselines. For an apple-to-apple comparison, our main experiments focus on LoRA-based merging methods (Pack of LLMs, Model Swarms, LoRAHub), which operate under the same adapter regime and are directly comparable. We will make this distinction—full-parameter coefficient evolution vs. adapter-level evolution—explicit in the revised manuscript.
>
> [1] Ruan W, Yang T, Zhou Y, et al. From Task-Specific Models to Unified Systems: A Review of Model Merging Approaches[J]. arXiv preprint arXiv:2503.08998, 2025.

---

> > ### Comment · Reviewer_zGDP · 2025-11-19
> >
> > Thank you for the clarifications, some of my concerns were partially addressed. I agree that full-parameter and adapter-based merging follow different traditions and serve different practical constraints. However, while I understand the authors’ point that
> >
> > > *“Full-parameter vs. adapter-based merging are different tracks in the literature. Our method, GENOME(+), is explicitly designed for adapter-based model merging…”*
> >
> > the distinction does not entirely eliminate the relevance of comparisons to prior evolutionary merging frameworks. Although LoRA imposes constraints on parameterization and compute, the underlying optimization problem: searching over combinations of task vectors or parameter deltas is structurally the same. Optimizing the merging coefficients of LoRA adapters is mathematically equivalent to optimizing the weighted combination of full-rank task vectors under a low-rank parameterization. This is why the operation you describe as “GENOME for LoRA adapters” maps directly onto the formulation used in evolutionary full-rank merging.
> >
> >
> > > *“The reviewer notes that we do not compare against Evolutionary Model Merging and MERGE³. We agree that there is conceptual overlap in the goal (evolutionary model merging), and we will explicitly discuss these works in the revised related work section.”*
> >
> > I appreciate the acknowledgment, but I want to clarify that the overlap is deeper than conceptual similarity of goals. GENOME is functionally equivalent to applying Evolutionary Model Merging in the LoRA setting, with the core loop—population-based search over merging weights evaluated via a small validation set—directly corresponding to the evolutionary formulation used in full-parameter methods. In other words, these methods operate on the same underlying search space structure, differing primarily in parameterization (full-rank vs. LoRA), not in optimization principle.
> >
> > For this reason, explicitly positioning GENOME relative to Evolutionary Model Merging and MERGE³ is important not only for completeness, but because GENOME represents a straightforward translation of these evolutionary strategies into the LoRA domain. Comparing or at least discussing them clarifies what is genuinely new in GENOME(+) and what is inherited from prior evolutionary merging techniques.

---

> ### Author Response · Authors · 2025-11-18
> **(2/2) About Weakness**
>
> > w3: GENOME+ shows major improvements but with higher computational overhead, not quantified beyond wall-clock time.
>
> We thank the reviewer both for pointing out the missing compute analysis and for acknowledging the strong performance of GENOME+. We agree that going beyond wall-clock time is important, and we will add an approximate FLOPs estimate for all methods, so that the computational overhead of GENOME+ is explicitly quantified as requested.
>
> > w4: “Succession” is effectively a simplified PSO update; not fully original.
>
> We agree that our succession mechanism is inspired by the update rule in particle swarm optimization, and we already state this connection in the paper. However, our contribution does not rely on succession alone: GENOME(+) combines selection, crossover, mutation, succession, and ensembling in a unified genetic framework operating directly on LoRA adapters. Integrating a PSO-style guidance term into this GA pipeline and demonstrating its effectiveness for evolutionary model merging in the adapter setting is, in our view, a contribution. We will slightly adjust the wording to make this “PSO-inspired” nature clearer, while also clarifying that succession is one component of the overall design rather than the sole source of novelty.

---

> ### Author Response · Authors · 2025-11-18
> **About Questions**
>
> > Q3: Can you provide FLOP-level estimates of GENOME+ vs. GENOME (given that GENOME+ introduces additional experience and ensemble operations)?
>
> We thank the reviewer for this suggestion. Below we provide approximate FLOP-level and time-level estimates under a unified setting.
>
> ## 1. FLOPs: GENOME vs. GENOME+ (LoRA setting)
>
> Shared assumptions (Gemma-2-2B-it, single task):
> - Model: Gemma-2-2B-it, per-token FLOPs: $ F_\text{token} \approx 1.1 \times 10^9 FLOPs/token $
> - Avg. tokens per call: C (input tokens) + S (output tokens) = 500 + 100 = 600
> → FLOPs per call $ F_\text{call} \approx 6.64 \times 10^{11} $
> - Data: 200 validation + 1000 test samples per task
> - Population N = 10, generations = 10
> - Total inference calls (init + 10 generations + test): 32,000 calls per method
> - LoRA params: $ P_\text{LoRA} \approx 1.1 \times 10^7 $ → one LoRA merge ≈ 2.2 $ \times 10^7 FLOPs $
> - Same evaluation budget for GENOME and GENOME+.
>
> Under these assumptions:
>
> | Method    | #Inference Calls | Inference FLOPs (≈) | #LoRA Merges | Merge FLOPs (≈) | Total FLOPs (≈)      |
> |-----------|------------------|---------------------|--------------|-----------------|----------------------|
> | GENOME    | 32,000           | 2.12 × 10^16        | 210          | 4.6 × 10^9      | 2.12 × 10^16         |
> | GENOME+   | 32,000           | 2.12 × 10^16        | 220          | 4.8 × 10^9      | 2.12 × 10^16         |
>
> GENOME+ uses the same number of model evaluations as GENOME; the extra experience (succession) update and ensemble are cheap LoRA merge operations on top of this budget. At the FLOP level, the difference between GENOME and GENOME+ is therefore negligible compared to the shared LLM inference cost.
>
> ⸻
>
> ## 2. Runtime illustration: LoRA vs. hypothetical full-parameter merging
> In addition, since several reviewers raised full-parameter merging methods, we also provide a simple runtime illustration comparing **LoRA-based** versus **hypothetical full-parameter** merging under the same setting.
>
> Under the same setting and a single RTX 4090, assuming no parallelism (serial decoding, which overestimates inference time):
> - Approx. decode time per call: 0.25 s (100 output tokens at ~400 tok/s)
> - Total inference calls: 32,000
>
> | Scenario                   | Decode Time (≈) | Merge Time (≈)      | Model Load Time (≈)             | Total Time (≈)   |
> |---------------------------|-----------------|----------------------|----------------------------------|------------------|
> | GENOME, LoRA merging      | 8,000 s         | 0.01 s              | ~20 s (one base model load)     | ~8,020 s         |
> | GENOME, full-param merging| 8,000 s         | ~3.3 s (210 merges)  | ~4,200 s (210 loads × 20 s/load) | ~12,200 s        |
>
> These numbers deliberately ignore batching and parallelism, so the decode times are conservative upper bounds. In a realistic system, inference can be parallelized and amortized, but repeated full-parameter model load/unload remains hard to hide and quickly becomes the dominant bottleneck. By contrast, LoRA merging keeps the base model resident and only touches small adapters, so its incremental cost is effectively negligible.
>
> We hope that the time-level comparisons make clear that frequent online evolution with many candidate evaluations is naturally aligned with LoRA-based merging, whereas full-parameter merging is far better suited to offline, one-shot combination. In this sense, the two belong to different practical “tracks”, and it is reasonable that our work focuses on the adapter-based setting.

---

> ### Author Response · Authors · 2025-11-21
> **About MERGE³ baseline**
>
> Given that several reviewers explicitly requested a comparison to MERGE³, we have added a corresponding baseline. Since MERGE³ is originally designed for full-parameter model merging and multilingual transfer, we implemented a LoRA-based variant, which we call MERGE3-LoRA, using the same optimization stack as the original code (i.e., pymoo with NSGA-II). The implementation has been pushed to our public codebase for transparency and reproducibility.
>
> We would also like to clarify our view on the relationship between GENOME(+) and MERGE³. We respectfully do not fully agree with the statement that “the overlap is deeper than conceptual similarity of goals” for GENOME(+) vs. MERGE³. In our opinion, LoRAHub and MERGE³ are more closely aligned in spirit: both apply existing generic optimizers to search over discrete or low-dimensional combination coefficients between models or layers—MERGE³ uses NSGA-II via pymoo, while LoRAHub uses Nevergrad for black-box optimization. By contrast, GENOME(+) defines LLM-oriented genetic operators that act directly on the adapter parameters themselves, enabling fine-grained crossover, mutation, selection, succession, and ensembling in LoRA weight space, rather than only optimizing mixing coefficients over a fixed set of experts.
>
> For fairness, all MERGE3-LoRA experiments are run under exactly the same setup as the other baselines: same datasets and splits, the same 5 random seeds (41, 42, 47, 53, 3407), and the same hardware. We report mean ± standard deviation over these 5 runs, consistent with the statistics we now provide for the other methods.
>
> We show the additional MERGE3-LoRA results below:
>
> | Method        | MMLU         | MMLUPro      | GSM8K        | MATH         | MGSM         | Flores-37    | ARC_C        | CSQA         | BBH          | DROP         | EmoryNLP     | MBPP         |
> |--------------|--------------|--------------|--------------|--------------|--------------|--------------|--------------|--------------|--------------|--------------|--------------|--------------|
> | LoRAHub      | 53.00±0.78   | 27.17±1.57   | 46.47±3.35   | 14.90±0.89   | 34.70±2.80   | 21.83±1.12   | 69.97±3.58   | 66.10±1.01   | 39.30±3.13   | 35.32±5.59   | 31.53±0.89   | 42.63±1.79   |
> | *MERGE³ (lora)* |   *53.16±0.70* | *28.09±2.08* | *48.03±0.84* | *14.90±1.57* | *37.81±5.20* | *22.19±2.00* | *70.01±1.37* | *67.68±4.53* | *39.40±2.40* | *33.28±4.68* | *32.77±2.09* | *42.48±1.03* |
> | Pack of LLMs | 54.30±2.46   | 27.67±0.78   | 38.13±1.01   | 11.50±2.91   | 31.06±1.12   | 19.24±2.35   | 70.22±1.12   | 63.30±3.13   | 38.90±0.89   | 21.30±4.25   | 30.46±2.80   | 42.56±1.01   |
> | Model Swarms | 55.91±1.07   | 27.77±1.69   | 45.82±2.49   | 15.06±2.38   | 33.15±2.70   | 21.76±2.57   | 68.53±2.94   | 68.76±2.09   | 38.80±3.23   | 34.52±5.90   | 33.98±2.91   | 42.56±2.65   |
> | GENOME       | **56.66±0.93**   | 27.77±2.20   | 49.34±2.04   | 15.72±1.66   | 37.19±1.79   | 22.78±1.91   | 70.69±2.50   | 68.22±2.71   | 40.00±2.10   | 35.28±6.31   | 34.97±2.51   | **43.60±1.86**   |
> | GENOME+      | 56.44±0.63   | **30.98±1.01**   | **51.24±1.34**   | **16.41±1.32**   | **39.55±1.57**   | **23.38±1.24**   | **74.38±1.45**   | **69.89±1.12**   | **41.10±1.57**   | **47.06±5.37**   | **38.81±1.45**   | 43.54±1.34   |
>
> Empirically, we observe that MERGE3-LoRA performs close to LoRAHub, which is consistent with their methodological similarity (both optimize discrete combination coefficients), and it remains clearly behind GENOME and GENOME+ on our 12 datasets. This aligns with our intuition: MERGE³ was designed with full-parameter merging in mind, where the large redundancy in full weight tensors allows discrete mixture recipes to work well. For LoRA adapters, which are much smaller and more structured, we believe that finer-grained parameter-level evolution (as in GENOME/GENOME+) is more suitable.
>
> In the revised manuscript, we will add these MERGE3-LoRA results in the appendix, clearly labeling them as our LoRA adaptation of MERGE³.
> We hope that these additional MERGE3-LoRA results help alleviate the reviewer’s concerns about missing baselines in this direction. If there are further aspects you would like us to clarify, we would be very happy to address them in the revision.

---

> ### Author Response · Authors · 2025-11-26
>
> We sincerely thank the reviewer’s comments. In the revised manuscript, we have (i) expanded the related work section to explicitly cite MERGE³ and to clearly state that our method is developed in the LoRA-based model-merging setting; and (ii) added MERGE³-LoRA as an additional baseline in the main experiments for a more comprehensive comparison. In addition, in this rebuttal we provide FLOP-level estimates for GENOME(+) and a simple time-based comparison, to quantitatively clarify the practical differences between full-parameter and LoRA-based merging in our setting.
>
> We would greatly appreciate it if you could let us know whether these revisions satisfactorily address your key concerns; if not, we are very happy to further clarify or refine the manuscript. If they do, we kindly ask you to consider updating your overall score in light of these clarifications and additional results.

---

> > ### Comment · Reviewer_zGDP · 2025-11-26
> >
> > I appreciate your efforts for adding an additional comparison, and I have raised my score accordingly. My concerns about the originality of the method have remained, since GENOME can be trivially derived from Evolutionary Model Merging in the LoRA parameter space, and MERGE3 seems to also offer GA through pymoo in its official codebase.
> > GENOME+ is more appealing, since it improves results by introducing a succession step that doesn't add much computational burden. But, this is also inherited from an existing technique (Model Swarms), so the novelty issue persists.

---

> ### Author Response · Authors · 2025-12-02
> **(1/2) Distinguishing GENOME from MERGE³ and Clarifying Novelty of GENOME+**
>
> Although we view full‑parameter model merging and adapter‑based (LoRA) merging as distinct technical lines, we implemented a LoRA variant of MERGE³ (MERGE3‑LoRA) and evaluated it on all 12 datasets under exactly the same splits, seeds, and compute as our methods. GENOME and GENOME+ consistently outperform MERGE3‑LoRA in this shared LoRA setting, so the remaining concern is not about missing baselines but about whether GENOME is essentially MERGE³ and therefore lacks novelty. The rest of this response clarifies—using explicit notation—how GENOME differs from MERGE³ in representation, operators, and search space, and then explains the additional contribution of GENOME+.
>
> **Key takeaways:**
> - MERGE³ evolves low-dimensional merge recipes $\lambda$, whereas GENOME evolves full LoRA adapters $w$ as individuals.
> - MERGE³’s crossover and mutation operate purely in recipe space, while GENOME’s operators act directly in the LoRA weight space.
> - Even when both are instantiated in the same LoRA parameter space (MERGE³‑LoRA vs. GENOME), MERGE³ explores at most a $p$‑dimensional surface $\Phi(\Lambda)$, while GENOME’s evolution in LoRA space reaches a much richer set; GENOME is therefore not a trivial reparameterization.
>
> 1. What the GA is evolving
>
> In MERGE³ , the evolutionary algorithm acts on a low-dimensional merge‑recipe vector. Given endpoint models
> $\{w_i\}_{i=1}^n$, MERGE³ assumes merged models of the form
>
> $$ w_{\tilde m} = \mathrm{Merge}(\{w_i\}_{i=1}^n; \lambda ), $$
> where $\mathrm{Merge}$ is a fixed merging operator (e.g., TIES, DARE, SLERP) and $ \lambda $ collects scalar hyperparameters such as interpolation weights and scaling factors. The GA (via pymoo) optimizes $ \lambda $ but never manipulates model parameters directly, treating merging as black‑box optimization over this recipe space.
>
> GENOME instead treats each individual as a full LoRA adapter
> $ w \in \mathbb{R}^d $,
> obtained by flattening all LoRA matrices attached to a shared base. The genotype is the LoRA weights themselves; evolutionary operators act directly on $ w $.
>
> Thus, to ``derive GENOME from MERGE³'' one would need to change (i) the genotype (from recipes to parameters), (ii) the operators (from recipe‑space SBX / polynomial mutation to LoRA‑space operators), and (iii) how offspring models are instantiated—far beyond a trivial configuration change.
>
> 2. Crossover and mutation: recipe space vs. LoRA space
>
> In MERGE³ / Mergenetic, both crossover and mutation are implemented on the recipe vector $ \lambda $:
> - Crossover: simulated binary crossover (SBX) combines parent recipes,
> $ \lambda^{(\text{child})} = \mathrm{SBX}(\lambda^{(a)}, \lambda^{(b)}) $,
> - Mutation: polynomial mutation perturbs each coordinate of $ \lambda $.
>
> The actual model parameters are then obtained in one step by applying the fixed operator $ \mathrm{Merge}(\cdot;\lambda^{(\text{child})}) $. The GA does not define how parameters are combined; it only chooses inputs $ \lambda $ to this external merge routine.
>
> In GENOME, the operators act directly in the LoRA parameter space. Given two parent adapters $w^{(a)}, w^{(b)} $, crossover produces
> $ w^{(\text{child})} = \tilde{\lambda}\, w^{(a)} + (1-\tilde{\lambda})\, w^{(b)} $,
> where $ \tilde{\lambda} $ is derived from the parents' fitness. Mutation is likewise defined on LoRA parameters via masked Gaussian perturbations:
> $$ w' = w + m \odot \varepsilon,\quad \varepsilon \sim \mathcal{N}(0, \sigma^2 I),\quad m \in \{0,1\}^d .$$
> These are weight‑space operators on LoRA tensors, not recipe‑space operators on $ \lambda $.
>
> Summarizing, MERGE³ evolves recipe vectors $ \lambda $ and then applies a fixed $ \mathrm{Merge} $ once to obtain parameters, whereas GENOME treats  $ \{w_i\} $ as genes and defines crossover/mutation directly on LoRA weights. Conceptually, one could modify Mergenetic so that the genotype is the full weight vector and re‑implement GENOME’s operators in that space—but that would simply reproduce our algorithm, not something already present in MERGE³.

---

> ### Author Response · Authors · 2025-12-02
> **(2/2) Distinguishing GENOME from MERGE³ and Clarifying Novelty of GENOME+**
>
> 3. Different reachable model families
>
> With $n$ endpoints and a fixed merge operator $ \mathrm{Merge} $, MERGE³ explores
> $$ \mathcal{H}_{\text{MERGE}^3}
> = \{\mathrm{Merge}(\{w_i\}; \lambda)\;|\; \lambda \in \Lambda\},
> $$
> where $\Lambda$ is a low‑dimensional domain of recipe hyperparameters, so every offspring is of the form $\mathrm{Merge}(\{w_i\}; \lambda)$ for some $\lambda \in \Lambda$.
>
> GENOME, in contrast, applies multiple rounds of LoRA‑space crossover and mutation (and in GENOME+, succession) directly on $w$. After $T$ generations, the reachable set is
> $$
> \mathcal{H}_{\text{GENOME}}
> = \{ w^{(T)} \;|\; w^{(0)} \in \{w_i\},\;
> w^{(t+1)} = \mathcal{O}(w^{(t)}, \text{population history}) \}, $$
> where $\mathcal{O}$ denotes the LoRA‑space operators.
> For a fair search‑space comparison, we consider the same ambient parameter space for both methods. In our experiments, this corresponds to MERGE³ instantiated as MERGE3‑LoRA, where the endpoints $\{w_i\}$ and all outputs of $\mathrm{Merge}(\{w_i\};\lambda)$ are flattened LoRA adapters on a shared base model, just like GENOME’s individuals.
>
> Formally, MERGE³ optimizes over a low-dimensional recipe space $\Lambda \subset \mathbb{R}^p$ (with $p \ll d$, where $d$ is the dimension of this shared parameter space—LoRA parameter dimension in our experiments) via a fixed merge map $\Phi : \Lambda \to \mathbb{R}^d$, $w_{\tilde m}=\Phi(\lambda)$. Thus the set of reachable parameters $\mathcal{H}_{\text{MERGE}^3}=\Phi(\Lambda)$ lies on at most a $p$-dimensional surface inside the $d$-dimensional parameter space (LoRA weight space in our setting). In contrast, GENOME defines crossover, mutation, and (in GENOME+) succession directly in the full LoRA weight space, with Gaussian perturbations that act in all $d$ dimensions, so its reachable set is not confined to such a low-dimensional surface. Even at the purely mathematical level of which parameters can be reached, MERGE³ and GENOME therefore operate on fundamentally different search spaces, and GENOME is not a trivial reparameterization of MERGE³.
>
> 4. LoRA‑specific design vs. generic EA back‑end
>
> MERGE³ / Mergenetic uses a generic pymoo GA that only sees recipe vectors and scalar fitness; LoRA is handled outside the EA by feeding LoRA checkpoints into fixed merge operators and evolving their hyperparameters. GENOME, in contrast, is built around LoRA adapters as individuals: initialization, crossover, mutation and (in GENOME+) succession are all defined on LoRA tensors, and the evaluation loop targets multi‑task LLM benchmarks with a shared frozen base and small validation sets. Thus the overlap between MERGE³ and GENOME is only the high‑level idea of using an evolutionary algorithm in the merging pipeline; the concrete representation, operators, and reachable models are substantially different.
>
> 5. Relation to Model Swarms and GENOME+
>
> The reviewer notes that GENOME+ introduces a succession step inspired by Model Swarms and questions novelty. We agree that Model Swarms shows PSO‑style sharing of personal‑best and global‑best information can benefit LLM populations, and GENOME+ adopts this idea via a LoRA‑space succession operator: for each individual $i$, we maintain an experience vector $ e_i $ of the same dimensionality as the LoRA weights $w_i$, and at generation $t$ update
> $$ e_i^{(t+1)} = \lambda_g e_g + \lambda_c e_c + \lambda_w e_w + \lambda_s e_i^{(t)},\quad
> w_i^{(t+1)} = w_i^{(t)} + \eta_e e_i^{(t+1)} $$,
> where $e_g$, $e_c$, and $e_w$ encode the global‑best, current‑best, and global‑worst experience in the population history.
>
> Compared to Model Swarms:
> - Model Swarms optimizes full LLM experts via a PSO‑style update in weight space, without GA‑style crossover/mutation.
> - GENOME+ combines (i) GA‑based crossover and mutation directly in LoRA space (Sections 1–2), (ii) a PSO‑inspired succession operator on LoRA experience vectors, and (iii) a final ensemble over the evolved adapters.
>
> Empirically, adding succession and ensemble systematically improves over GENOME without succession and over existing model‑merging baselines under the same evaluation budget and low‑data regimes. Thus, while we fully acknowledge the PSO inspiration from Model Swarms, the LoRA‑space GA + succession + ensemble framework and its demonstrated gains are, to our knowledge, new and constitute the main contribution of GENOME+.

---

### Official Review · Reviewer_93BN · 2025-10-27

**Soundness:** 2
**Presentation:** 2
**Contribution:** 2
**Rating:** 2
**Confidence:** 4

**Summary:**

The paper introduces GENOME, a new evolutionary-based post-training techinque that extends ideas from evolutionary model merging. Model merging is a post-training strategy that constructs new models by combining previously fine-tuned models sharing a common pretrained backbone, allowing the reuse of specialized expertise across domains. In contrast, evolutionary algorithms are natural-evolution-inspired, gradient-free optimization methods that evolve candidate solutions through crossover, mutation, and selection. GENOME unifies these two paradigms by treating each fine-tuned model as an individual in a population whose parameters evolve toward higher task fitness via biologically inspired mechanisms (crossover → simple merging, mutation → adding noise). Building upon this foundation, GENOME+ enriches the process with succession,  a mechanism for transferring knowledge from high-performing models to others via another merging step, and ensemble inference, which aggregates outputs from the top-performing individuals for more robust predictions via an ensemble. The framework is evaluated using the Gemma-2-2b-it model across twelve diverse datasets, demonstrating consistent performance gains (even though, it is hard to judge them significant due to lack of std report). However, the paper fail to acknowledge more recent methods in model merging and evolutionary optimization, therefore omitting crucial baselines for data-intensive merging pipelines.

**Strengths:**

- **Extensive Evaluation:** The paper presents an impressively thorough empirical study of the proposed merging method on **Gemma-2-2b-it**, covering a wide range of datasets and evaluation settings. Such comprehensive testing is rare in the model merging literature, where evaluations are often limited or approximate. The authors deserve particular credit for this, despite making the significance of these findings is tricky to interpret without reporting the standard deviation.
    - I particularly appreciated the ablation, it is very indepth. Nevertheless, it has the same problem that I mentioned before (its lack of standard deviation reporting remains a limitation for understanding its actual validity)
- **Original Evolutionary Steps:** The **succession** and **ensemble** components introduced in **GENOME+** represent genuine novelties compared to prior model merging methods, enabling knowledge transfer and collective inference within the evolving model population. Nevertheless, the broader evolutionary framework itself is not conceptually new, as it closely parallels established approaches such as [1, 2], and mainly adapts well-known genetic algorithm principles to the context of large language model merging rather than introducing a fundamentally new post-training optimization paradigm.
- **Relevant Results:** The paper demonstrates consistent improvements over **Model Swarms**, with GENOME+ achieving higher average performance across multiple benchmarks and task types. This suggests a meaningful advancement in population-based model adaptation. However, because the results are reported only as averages across random seeds without standard deviations, it is difficult to fully assess the reliability or statistical strength of these gains. Providing variance estimates would have made the comparison more convincing and transparent. Furthemore, it completely miss baseline against other SOTA method such as Akiba or MERGE3 [1,2].

- *The code* is well done and it is a contribution

[1] Akiba, Takuya, et al. "Evolutionary optimization of model merging recipes." *Nature Machine Intelligence* (2025): 1-10.
[2] Mencattini, Tommaso, et al. "MERGE3: Efficient Evolutionary Merging on Consumer-grade GPUs." Proceedings of The Forty-Second International Conference on Machine Learning (ICML) (2025).

**Weaknesses:**

- **Presentation quality.** The paper’s presentation could be improved, as the overall layout, and particularly the teaser in **Figure 1,** appears overly busy and difficult to follow.
    - Moreover, the way performance gains are reported can be somewhat misleading: for instance, the claimed **10.75% improvement** over Model Swarms is expressed as a *relative* rather than *absolute* increase, which may exaggerate the perceived impact. *Given that the true gain is likely within the single-digit range, statistical testing or at least reporting standard deviations would be necessary to judge the real significance of these improvements.*
- **Overstated Novelty.** One of the paper’s main claims is the introduction of an **LLM population-based evolution** framework. However, this concept is **not novel**. While Akiba et al. [1] may have offered a narrower perspective (focusing e.g. on CMAES), similar ideas of LLM evolution through model merging have already been explored, notably in **MERGE3 [2]**, which the authors do not adequately acknowledge. This omission hides a reduced originality of the contribution: the paper does not truly introduce the first formulation of population-based LLM evolution, but rather a **variation of previously established approaches**. Although the inclusion of an ensemble step arguably makes the method somewhat more general than prior evolutionary model merging frameworks, its specific role and added value remain unclear (see below). Moreover, the other three evolutionary operations, crossover, mutation, and selection, are simply merging step with gaussian noise, so they are not a true novel contribution to evolutionary merging of LLMs.
- **Poor Baseline Selection:** A major limitation of the paper is the **lack of appropriate comparison**, which raises concerns about the validity of the reported improvements. Because key prior work is not fully considered, the evaluation omits comparisons with the most relevant methods, most notably **MERGE3**, which also implements an evolutionary merging framework compatible with consumer-grade GPUs, and **Akiba et al.**, whose approach shares conceptual overlap with GENOME+.
    - Combined with the **absence of standard deviations** in performance reporting and therefore the unclear magnitude of gains over **Model Swarms**, it becomes difficult to assess whether the observed improvements are statistically or practically meaningful.
    - Furthermore, the experiments rely exclusively on **ad hoc fine-tuned experts** rather than publicly available or standardized fine-tuning checkpoints (e.g., from Hugging Face), further weakening the robustness and comparability of the evaluation. As a result, the current experimental setup does not convincingly demonstrate a clear or generalizable advantage over existing baselines.
- **Result Significance:** A broader issue throughout the paper concerns the **lack of statistical significance analysis and standard deviation reporting**, which undermines confidence in the reported results. Almost all performance comparisons are presented as mean values without any measure of variance, making it difficult to assess whether observed differences are consistent or simply due to random fluctuations. This limitation affects both the main results and the ablation studies, where conclusions about the contribution of individual components may not be statistically reliable. Incorporating statistical significance tests (e.g., t-tests or confidence intervals) **or at least standard deviation** would be essential to substantiate the claimed improvements and to better demonstrate the robustness and reproducibility of the proposed approach.

In conclusion, the work presents an interesting framework: I found GENOME+ definitely interesting, and I would like to try in the future. But I suggest **rejecting** because the claimed novelty is overstated relative to prior evolutionary model-merging, which are ignored. Without that, the paper proposes a method that builds on others, and therefore should show clear improvement. Nevertheless, the empirical validation lacks key baselines, variance/significance reporting, and clear evidence that gains are reliable or generalizable.

**Questions:**

Q1) I do not fully understand how the ensemble mechanism operates. Is the ensembled model subsequently considered as part of the population? If so, how is this “ensembled individual” integrated into the standard GENOME operations, such as mutation and crossover?

Q2) To fairly compare data-free merging methods with data-based merging approaches, it would be necessary to include a baseline using a random or Bayesian hyperparameter search. Specifically, have you conducted an equivalent FLOP-budgeted random or Bayesian search for DARE_TIES or similar baselines to ensure a fair comparison?

---

> ### Author Response · Authors · 2025-11-18
> **(1/2) About Weakness**
>
> > Response to w1 (performance gains and variance reporting)
>
> We appreciate the reviewer’s concern about potentially misleading reporting of performance gains. We would like to clarify that we do not intend to exaggerate the improvements. All experiments in the paper—including those for our baselines—are run with **five different random seeds**, and the reported numbers are the mean performance over these runs. The relative improvement percentages are computed uniformly from these mean scores.
>
> We also note that the main baselines we compare against (e.g., **LoRAHub, Pack of LLMs, Model Swarms**) do not report standard deviations or statistical significance tests in their original papers. Our current presentation is therefore aligned with the prevailing practice in this line of work.
>
> **To address the reviewers' concerns, we will supplement this data at soon.**
>
> > Response to w2 (novelty and relation to prior evolutionary methods)
>
> What we aim to emphasize is that these methods (LoraHub, Merge3) largely **treat model merging as a coefficient optimization problem** over a small number of mixture weights (often at the layer or model level), typically framed as a discrete or low-dimensional continuous optimization task (e.g., via NSGA-II, pymoo, or similar). In contrast, **GENOME(+) defines and implements genetic operators directly over the adapter parameters themselves** (specifically LoRA adapters). This allows us to perform crossover and mutation at a much **finer granularity—down to the parameter / matrix level of the adapters—rather than only tuning sparse combination coefficients across experts.**
>
> > Response to w4 (use of custom fine-tuned experts vs. public checkpoints)
>
> We acknowledge the reviewer’s concern about using ad hoc fine-tuned experts. However, this choice is largely **necessitated by the problem setting**. LoRA-based merging methods require that the adapters share **the same base model and compatible adapter dimensions / parameterization**. Publicly available checkpoints rarely satisfy these constraints simultaneously (same base model, same LoRA rank, same fine-tuning setting), making it difficult to construct a fair and controlled comparison using off-the-shelf models.
>
> Our setup follows the precedent of prior work such as **Model Swarms**, where all experts are fine-tuned under a unified recipe to ensure compatibility and fairness. Importantly, we **open-source all our fine-tuned experts**, so that anyone can reproduce the entire optimization process on a consumer-grade GPU starting from exactly the same experts.
>
> We therefore view the use of a **consistent set of fine-tuned experts on a shared base model** as a fair and practical starting point for evaluating LoRA-based merging methods, rather than a weakness.

---

> ### Author Response · Authors · 2025-11-18
> **(2/2)  About weakness**
>
> > Response to Q3 (baseline selection and comparison with full-parameter methods)
>
> Our work, GENOME(+), is specifically focused on **adapter-based model merging**, concretely on **LoRA-based adapters**. Accordingly, we selected baselines that are also **LoRA-based merging methods**, such as Pack of LLMs, Model Swarms, and LoRAHub, so that all methods operate in the **same parameterization and problem setting**. The methods highlighted by the reviewer, such as **MERGE3** and **Evolutionary Optimization of LLMs**, primarily target **full-parameter model merging**, and are evaluated mainly on **multilingual transfer**.
>
> In recent taxonomies of model merging, full-parameter merging and LoRA-based (adapter-based) merging are explicitly categorized as separate classes, since LoRA models impose different computational and compatibility constraints and often require dedicated merging techniques [1].
> For this reason, we believe that directly comparing our LoRA-merging framework against full-parameter merging methods—especially those tailored to multilingual scenarios—would mix different problem settings.
>
> Moreover, these full-parameter methods have so far been primarily demonstrated on multilingual benchmarks, without peer-reviewed evidence that they are effective across **multiple domains and many datasets**. By contrast, GENOME(+) is evaluated on **12 datasets across 5 domains**. From both the **object of study** (adapters vs. full weights) and the **experimental focus**, we therefore believe that our current baseline selection is reasonable for the scope of this work.
>
> From a systems and deployment perspective, iterative LoRA-based merging tends to be more practical than full-parameter merging in current LLM serving stacks. Modern high-performance inference backends such as vLLM and SGLang, as well as several recent multi-LoRA or multi-tenant serving systems, natively support attaching and switching multiple LoRA adapters on top of a single base model. For example, vLLM allows specifying LoRA adapters on a per-request basis and provides mechanisms for loading and unloading adapters at runtime, and works such as LoRA-Inlaid [2] and dLoRA [3] further demonstrate that many LoRA adapters can be scheduled online over a shared (sometimes quantized) base model while maintaining low latency and high throughput. In this setting, frameworks like GENOME(+) can keep the base model resident in memory and only compose or swap relatively small adapters during multi-round evolution, which makes large-scale search feasible even on commodity GPUs.
>
> By contrast, full-parameter merging typically operates by directly modifying the entire set of model weights. Existing full-parameter merging methods are usually designed as offline or training-time procedures: task vectors are combined once to produce a new checkpoint, or merging is integrated into the optimizer during RLHF/alignment (e.g., Online Merging Optimizers [4]), after which the resulting model is deployed as a static checkpoint rather than being re-merged on a per-request basis in the serving stack. Incorporating such full-parameter merging into a multi-round online search loop would generally require repeatedly materializing merged weights and reloading the full model into the inference backend, so the wall-clock cost becomes dominated not only by evaluation but also by frequent load/unload overhead.
>
> Given this engineering context, our focus on adapter-based (LoRA) merging is not only conceptually aligned with our problem setting (evolving adapters under a shared base model), but also better matches the capabilities and constraints of current LLM serving infrastructures. This provides an additional rationale for prioritizing LoRA-based baselines in our experimental comparison, rather than placing our method directly alongside full-parameter merging approaches that target a different deployment regime.
>
> [1] Ruan W, Yang T, Zhou Y, et al. From Task-Specific Models to Unified Systems: A Review of Model Merging Approaches[J]. arXiv preprint arXiv:2503.08998, 2025.
>
> [2] Xia Y, Fu F, Zhang W, et al. Efficient multi-task llm quantization and serving for multiple lora adapters[J]. Advances in Neural Information Processing Systems, 2024, 37: 63686-63714.
>
> [3] Gao C, Zhang S Q. Dlora: Distributed parameter-efficient fine-tuning solution for large language model[J]. arXiv preprint arXiv:2404.05182, 2024.
>
> [4] Lu K, Yu B, Huang F, et al. Online merging optimizers for boosting rewards and mitigating tax in alignment[J]. arXiv preprint arXiv:2405.17931, 2024.

---

> ### Author Response · Authors · 2025-11-18
> **About Questions**
>
> > Q1) I do not fully understand how the ensemble mechanism operates. Is the ensembled model subsequently considered as part of the population? If so, how is this “ensembled individual” integrated into the standard GENOME operations, such as mutation and crossover?
>
> Our ensemble mechanism is not injected back into the evolutionary population as a new “individual.” Instead, it is a purely evaluation-time operation built on top of the current population.
>
> Concretely, GENOME’s standard operators (crossover, mutation, selection) always act on single or more models / individuals represented by their adapter parameters. During evolution, the population consists only of such individuals, and all genetic operators are defined directly on these adapters.
>
> The ensemble step is applied after we have a set of evolved individuals: at evaluation/inference time, we take a small subset of high-performing individuals (e.g., top-k models) from the current population and perform a voting over their outputs. This improves robustness and performance, but it does not alter the underlying evolutionary dynamics, and the ensembled predictor itself is not used as a parent for further crossover or mutation in our experiments.
>
> This decoupling is intentional and reflects a design goal of GENOME(+): because we re-implement the operators directly for the LLM–adapter setting, we are not tied to a specific off-the-shelf optimizer (such as NSGA-II or pymoo) and can treat components like “evolution over adapters” and “prediction-time ensembling” as modular, plug-and-play parts.
>
> > Q2) To fairly compare data-free merging methods with data-based merging approaches, it would be necessary to include a baseline using a random or Bayesian hyperparameter search. Specifically, have you conducted an equivalent FLOP-budgeted random or Bayesian search for DARE_TIES or similar baselines to ensure a fair comparison?
>
> First, regarding data-free merging, we would like to clarify that our Expert Fusion baseline in Tables 1–4 is indeed a data-free merging method: it directly averages the LoRA vectors of the experts without using any target-task data to tune coefficients. In that sense, it provides a simple but representative data-free baseline within our adapter-based setting.
>
> Second, we did experiment with a Bayesian-search-style baseline during preliminary internal testing. However, as discussed earlier in the paper, GENOME(+) operates at a much finer parameter granularity than methods that only optimize a small number of mixture or layer-wise coefficients (e.g., as in some prior merging frameworks). Our operators act directly on the adapter parameters themselves, rather than on a low-dimensional set of scalar combination weights.
>
> In this high-dimensional regime, we found that Bayesian optimization converged very slowly under a realistic FLOP budget and could not match the performance of population-based evolution within a comparable compute envelope. In other words, when trying to apply random/Bayesian search at the same level of granularity as GENOME(+), the search became inefficient and did not yield competitive models in the same time. For this reason, and to keep the experimental section focused and computationally feasible, we decided not to include these preliminary Bayesian-search results in the main paper.
>
> If the reviewers consider it important, we are happy to add a brief discussion or an additional appendix table summarizing our preliminary Bayesian-search experiments and their convergence behavior.

---

> ### Author Response · Authors · 2025-11-19
> **About absence of standard deviations**
>
> We thank the reviewer for repeatedly pointing out the absence of standard deviations in our main results. To fully address this concern, we have now computed and added per-dataset standard deviations over five random seeds (41, 42, 47, 53, 3407) for all main methods.
>
> Our initial decision to report only means in the main tables was not an attempt to hide variance, but a presentation choice: the main experiment covers 12 datasets × multiple methods, and adding “mean ± std” everywhere would make the tables extremely cluttered and hard to read. This style of reporting mean scores only is also consistent with prior work we compare to (e.g., LoRAHub [1], Pack of LLMs [2], Model Swarms [3]).
>
> To address the reviewers’ concerns while keeping the main text readable, we will include the full variance statistics in the appendix. For convenience, we reproduce them here:
>
> | Method        | MMLU         | MMLUPro      | GSM8K        | MATH         | MGSM         | Flores-37    | ARC_C        | CSQA         | BBH          | DROP         | EmoryNLP     | MBPP         |
> |--------------|--------------|--------------|--------------|--------------|--------------|--------------|--------------|--------------|--------------|--------------|--------------|--------------|
> | LoRAHub      | 53.00±0.78   | 27.17±1.57   | 46.47±3.35   | 14.90±0.89   | 34.70±2.80   | 21.83±1.12   | 69.97±3.58   | 66.10±1.01   | 39.30±3.13   | 35.32±5.59   | 31.53±0.89   | 42.63±1.79   |
> | Pack of LLMs | 54.30±2.46   | 27.67±0.78   | 38.13±1.01   | 11.50±2.91   | 31.06±1.12   | 19.24±2.35   | 70.22±1.12   | 63.30±3.13   | 38.90±0.89   | 21.30±4.25   | 30.46±2.80   | 42.56±1.01   |
> | Model Swarms | 55.91±1.07   | 27.77±1.69   | 45.82±2.49   | 15.06±2.38   | 33.15±2.70   | 21.76±2.57   | 68.53±2.94   | 68.76±2.09   | 38.80±3.23   | 34.52±5.90   | 33.98±2.91   | 42.56±2.65   |
> | GENOME       | 56.66±0.93   | 27.77±2.20   | 49.34±2.04   | 15.72±1.66   | 37.19±1.79   | 22.78±1.91   | 70.69±2.50   | 68.22±2.71   | 40.00±2.10   | 35.28±6.31   | 34.97±2.51   | 43.60±1.86   |
> | GENOME+      | 56.44±0.63   | 30.98±1.01   | 51.24±1.34   | 16.41±1.32   | 39.55±1.57   | 23.38±1.24   | 74.38±1.45   | 69.89±1.12   | 41.10±1.57   | 47.06±5.37   | 38.81±1.45   | 43.54±1.34   |
>
> From this table, we observe that GENOME+ (with ensemble) is typically **the most stable method** (smallest standard deviations across tasks), while GENOME and Model Swarms—both performing finer-grained, dynamic model merging—have similar variance and are less stable than GENOME+. LoRAHub and Pack of LLMs, which optimize discrete combination coefficients, tend to show higher variability.
>
> In the revised manuscript, we will:
> - Keep the main tables as means for readability, and
> - Add a table like the one above to the appendix, explicitly reporting standard deviations to fully address the reviewers’ concerns.
>
> [1] Huang C, Liu Q, Lin B Y, et al. Lorahub: Efficient cross-task generalization via dynamic lora composition[J]. arXiv preprint arXiv:2307.13269, 2023.
>
> [2] Mavromatis C, Karypis P, Karypis G. Pack of llms: Model fusion at test-time via perplexity optimization[J]. arXiv preprint arXiv:2404.11531, 2024.
>
> [3] Feng S, Wang Z, Wang Y, et al. Model swarms: Collaborative search to adapt llm experts via swarm intelligence[J]. arXiv preprint arXiv:2410.11163, 2024.

---

> ### Author Response · Authors · 2025-11-21
> **About MERGE³ baseline**
>
> Given that several reviewers explicitly requested a comparison to MERGE³, we have added a corresponding baseline. Since MERGE³ is originally designed for full-parameter model merging and multilingual transfer, we implemented a LoRA-based variant, which we call MERGE3-LoRA, using the same optimization stack as the original code (i.e., pymoo with NSGA-II). The implementation has been pushed to our public codebase for transparency and reproducibility.
>
> We would also like to clarify our view on the relationship between GENOME(+) and MERGE³. We respectfully do not fully agree with the statement that “the overlap is deeper than conceptual similarity of goals” for GENOME(+) vs. MERGE³. In our opinion, LoRAHub and MERGE³ are more closely aligned in spirit: both apply existing generic optimizers to search over discrete or low-dimensional combination coefficients between models or layers—MERGE³ uses NSGA-II via pymoo, while LoRAHub uses Nevergrad for black-box optimization. By contrast, GENOME(+) defines LLM-oriented genetic operators that act directly on the adapter parameters themselves, enabling fine-grained crossover, mutation, selection, succession, and ensembling in LoRA weight space, rather than only optimizing mixing coefficients over a fixed set of experts.
>
> For fairness, all MERGE3-LoRA experiments are run under exactly the same setup as the other baselines: same datasets and splits, the same 5 random seeds (41, 42, 47, 53, 3407), and the same hardware. We report mean ± standard deviation over these 5 runs, consistent with the statistics we now provide for the other methods.
>
> We show the additional MERGE3-LoRA results below:
>
> | Method        | MMLU         | MMLUPro      | GSM8K        | MATH         | MGSM         | Flores-37    | ARC_C        | CSQA         | BBH          | DROP         | EmoryNLP     | MBPP         |
> |--------------|--------------|--------------|--------------|--------------|--------------|--------------|--------------|--------------|--------------|--------------|--------------|--------------|
> | LoRAHub      | 53.00±0.78   | 27.17±1.57   | 46.47±3.35   | 14.90±0.89   | 34.70±2.80   | 21.83±1.12   | 69.97±3.58   | 66.10±1.01   | 39.30±3.13   | 35.32±5.59   | 31.53±0.89   | 42.63±1.79   |
> | *MERGE³ (lora)* |   *53.16±0.70* | *28.09±2.08* | *48.03±0.84* | *14.90±1.57* | *37.81±5.20* | *22.19±2.00* | *70.01±1.37* | *67.68±4.53* | *39.40±2.40* | *33.28±4.68* | *32.77±2.09* | *42.48±1.03* |
> | Pack of LLMs | 54.30±2.46   | 27.67±0.78   | 38.13±1.01   | 11.50±2.91   | 31.06±1.12   | 19.24±2.35   | 70.22±1.12   | 63.30±3.13   | 38.90±0.89   | 21.30±4.25   | 30.46±2.80   | 42.56±1.01   |
> | Model Swarms | 55.91±1.07   | 27.77±1.69   | 45.82±2.49   | 15.06±2.38   | 33.15±2.70   | 21.76±2.57   | 68.53±2.94   | 68.76±2.09   | 38.80±3.23   | 34.52±5.90   | 33.98±2.91   | 42.56±2.65   |
> | GENOME       | **56.66±0.93**   | 27.77±2.20   | 49.34±2.04   | 15.72±1.66   | 37.19±1.79   | 22.78±1.91   | 70.69±2.50   | 68.22±2.71   | 40.00±2.10   | 35.28±6.31   | 34.97±2.51   | **43.60±1.86**   |
> | GENOME+      | 56.44±0.63   | **30.98±1.01**   | **51.24±1.34**   | **16.41±1.32**   | **39.55±1.57**   | **23.38±1.24**   | **74.38±1.45**   | **69.89±1.12**   | **41.10±1.57**   | **47.06±5.37**   | **38.81±1.45**   | 43.54±1.34   |
>
> Empirically, we observe that MERGE3-LoRA performs close to LoRAHub, which is consistent with their methodological similarity (both optimize discrete combination coefficients), and it remains clearly behind GENOME and GENOME+ on our 12 datasets. This aligns with our intuition: MERGE³ was designed with full-parameter merging in mind, where the large redundancy in full weight tensors allows discrete mixture recipes to work well. For LoRA adapters, which are much smaller and more structured, we believe that finer-grained parameter-level evolution (as in GENOME/GENOME+) is more suitable.
>
> In the revised manuscript, we will add these MERGE3-LoRA results in the appendix, clearly labeling them as our LoRA adaptation of MERGE³.
> We hope that these additional MERGE3-LoRA results help alleviate the reviewer’s concerns about missing baselines in this direction. If there are further aspects you would like us to clarify, we would be very happy to address them in the revision.

---

> ### Author Response · Authors · 2025-11-26
>
> We sincerely thank the reviewer's detailed feedback. In the revised manuscript, we have (i) updated the abstract and introduction to explicitly mark all reported percentage gains as relative improvements, to avoid any ambiguity; (ii) expanded the related work section to explicitly cite MERGE³ and to clearly state that our method is developed in the LoRA-based model-merging setting; (iii) added MERGE³-LoRA as an additional baseline in the main experiments for a more comprehensive comparison; and (iv) included results with standard deviations for the main methods in the appendix. All of these changes are highlighted in blue for ease of inspection.
>
> We would greatly appreciate it if you could let us know whether these revisions satisfactorily address your key concerns; if not, we are very happy to further clarify or refine the manuscript. If they do, we kindly ask you to consider updating your overall score in light of these clarifications and additional results.

---

### Official Review · Reviewer_ZdLh · 2025-11-01

**Soundness:** 3
**Presentation:** 4
**Contribution:** 3
**Rating:** 6
**Confidence:** 3

**Summary:**

The paper formalizes LLM population-based evolution and proposes GENOME and GENOME+, which evolve populations of LoRA adapters using fitness-weighted crossover, masked Gaussian mutation, elite + fitness-proportional selection, and (for GENOME+) a succession (experience vector) update plus top-k ensembling at inference. Experiments across 12 datasets report notable gains over several dynamic model-merging baselines and claim scalability up to 40 models and feasibility on a single RTX 4090.

**Strengths:**

The paper addresses a timely, practical problem: reusing many fine-tuned LoRA experts without full gradient fine-tuning. As the performance of open source models improves, this provides a valuable direction for combining complementary strengths.

It relies on simple, implementable evolutionary operators (linear mixing, masked mutation, selection) that are attractive as a gradient-free approach.

Broad empirical ambition: diverse tasks (12 datasets), multi-task and zero-shot experiments, and population-scaling studies, base models of two different practical sizes.

The authors indicate code release and include many appendix details, which could support reproducibility and adoption.

**Weaknesses:**

In my opinion there are two missing critical baselines: no gradient-based (multi-task) LoRA fine-tuning on the same 200 samples and no simple top-k ensemble of original experts to separate ensemble gains from evolution.

It would be appreciated if the authors could improve on the presentation of key algorithmic details, e.g. the exact per-layer/parameter merging for LoRA adapters, mask sampling granularity (per-parameter vs per-matrix), mutation application, and successive normalization are unclear.

There is some risk of overfitting to the small 200-sample validation set used for evolution and selection; it would interesting to investigate the robustness to the specific validation choice and size is not demonstrated.

The compute and hyperparameter tuning details appear incomplete: how many tuning runs were used per method, were baselines given the same hyperparameter budget, and how many seeds were used?

Finally, all results consider two base models Gemma-2-2B-it, Llama-3.1-8B. In order to demonstrate general applicability, the authors should potentially look into larger base models, e.g. >10B, etc. Additionally, why were the Llama results not explicitly discussed in the main text?

**Questions:**

Can you provide direct baselines: (a) LoRA gradient-based fine-tuning on the same 200 validation samples with a comparable compute budget; (b) top-k ensemble of the original experts (no evolution); (c) naive averaging of top experts. Report these on main datasets with identical prompts and compute constraints.

Additionally, can you add low-level implementation details for parameter merging: how are LoRA matrices combined elementwise or per-layer? How are low-rank decompositions preserved?

Could you demonstrate robustness to validation set choice and size: repeat adaptation/evolution with multiple disjoint 200-example validation samples and with other sizes (e.g., 50, 500) and report stability of selection and final test performance.

Also, can you provide ablations that isolate ensemble vs evolution: (i) evolve without ensemble and report best individual performance; (ii) ensemble-only of original experts (no evolution); (iii) random merging + ensemble. Report statistical significance.

Can you please extend the analysis to one more large base model (>10B)?

Finally, how does this work relate and contrast to CycleQD ([1], Kuroki, et al., 2024) especially with regards to the multi-task setup?

[1] Kuroki, S., Nakamura, T., Akiba, T. and Tang, Y., 2024. Agent skill acquisition for large language models via cycleqd. arXiv preprint arXiv:2410.14735.

---

> ### Author Response · Authors · 2025-11-17
> **Q: Can you add low-level implementation details for parameter merging: how are LoRA matrices combined elementwise or per-layer? How are low-rank decompositions preserved?**
>
> > Since the reviewers raised multiple detailed points and our additional experiments are progressing at different speeds, we respond to the comments in a slightly reordered fashion, and we kindly ask for your understanding.
>
> We thank the reviewer for pointing this out. Below we clarify how LoRA parameters are merged and why the low-rank structure is preserved.
>
> ### (1) Representation and low-rank form
>
> For each adapted linear layer we use the standard LoRA parameterization
> $$\Delta W = \frac{\alpha}{r} A B^\top,$$
>
> where $ \(A \in \mathbb{R}^{d_\text{out} \times r}\) $, $\(B \in \mathbb{R}^{r \times d_\text{in}}\) $, and $ \(r\) $ is the (fixed) LoRA rank.
>
> Each individual stores a dictionary of LoRA tensors:
> ```python
> self.x: Dict[str, torch.Tensor]
> ```
> with keys such as
> - "model.layers.5.self_attn.v_proj.lora_A.weight"
> - "model.layers.5.self_attn.v_proj.lora_B.weight"
>
> pointing to the corresponding LoRA matrices (A) or (B) for each layer/module. All individuals share the same LoRA configuration, so the shapes (and thus rank (r)) are identical across parents.
>
> ### (2) How merging is done (per-layer, elementwise)
>
> Merging is performed per layer: parameters are matched by their key (layer/module name), and within each layer the corresponding LoRA matrices from different parents are combined elementwise.
>
> If (K) parents have LoRA matrices ($ {A_\ell^{(k)}}{k=1}^K $) for layer (\ell) and mixing coefficients ($ {\lambda_k}{k=1}^K $) (with ($ \sum_k \lambda_k = 1 $)), the child’s matrix is
> $$A_\ell^{(\text{child})} = \sum_{k=1}^{K} \lambda_k A_\ell^{(k)},$$
>
> and analogously for ($ B_\ell $). A simplified implementation is:
> ```python
> def merge_lora_weights(parent_weights, mixing_coeffs):
>     # parent_weights: list[Dict[str, Tensor]], mixing_coeffs: Tensor[K]
>     child_weights = {}
>     K = len(parent_weights)
>     for key in parent_weights[0].keys():
>         stacked = torch.stack([p[key] for p in parent_weights], dim=0)  # [K, *shape]
>         coeffs = mixing_coeffs.view(K, *([1] * (stacked.ndim - 1)))
>         merged = torch.sum(coeffs * stacked, dim=0)  # elementwise weighted sum
>         child_weights[key] = merged
>     return child_weights
> ```
>
> Thus, merging is per-layer (by key) and elementwise within each LoRA matrix.
>
> ### (3) Preservation of the low-rank decomposition
> - We never materialize the full ($ \Delta W $) or modify base weights; all operations (including the mutation operator in the code snippet) are applied directly to (A) and (B).
> - Merging and mutation only perform linear/affine operations on tensors with fixed shapes (($ d_\text{out}, r $)) and (($ r, d_\text{in} $)).
> Therefore, the LoRA rank (r) and the factorized form
> $ \Delta W’ = \frac{\alpha}{r} A’ {B’}^\top $
> are preserved for every layer and every individual.
>
> With this design, we can explore model merging at a finer granularity than LoRAHub, and Pack of LLMs.

---

> ### Author Response · Authors · 2025-11-17
> **Q: Can you provide direct baselines: (a) LoRA gradient-based fine-tuning on the same 200 validation samples with a comparable compute budget; (b) top-k ensemble of the original experts (no evolution); (c) naive averaging of top experts. Report these on main datasets with identical prompts and compute constraints.**
>
> Regarding the requested baselines, we would like to clarify our setup and how we address points (a)–(c).
>
> First, the benchmarks we use are standard LLM evaluation suites that only provide **inputs and final answers**, but generally *lack intermediate reasoning traces* (“thoughts”) and have heterogeneous output formats. Direct LoRA fine-tuning on such raw (input, answer) pairs is known to be weak supervision for reasoning-heavy tasks, and also leads to format-parsing issues at evaluation time. Nevertheless, to fully address the reviewer’s concern, we construct two fine-tuning datasets and run LoRA baselines on **Gemma-2-2B-it** under the **same training hyperparameters** as our original experts:
> 1. **Raw benchmark fine-tuning**: we fine-tune Gemma-2-2B-it directly on the original (question, answer) benchmark pairs (200 validation samples per task).
>
> 2. **Distilled benchmark fine-tuning**: we use a powerful model (GPT-5) to (i) produce intermediate reasoning and (ii) normalize answer formats, and then fine-tune Gemma-2-2B-it on this enhanced dataset (i.e., this baseline *adds* extra supervision from GPT-5, whereas our method and all model-merging baselines do **not** use any external teacher model or additional supervision).
>
> For **(b) “top-k ensemble of the original experts (no evolution)”**, we explicitly report a **top-3 ensemble** of the original experts, matching the main text setup (same \(k\), same prompts).
>
> For **(c) “naive averaging of top experts”**, this is exactly our **expert fusion** baseline already reported in Tables 1–4.
>
> Since these additional baselines do not involve evolutionary operation, we report them under a single seed (seed = 42). The corresponding results are summarized in the under.
>
> | Setting                         | mmlu  | mmlupro | gsm8k | math  | mgsm  | flores37 | arcc  | csqa  | bbh   | drop  | emorynlp | mbpp  |
> |---------------------------------|-------|---------|-------|-------|-------|----------|-------|-------|-------|-------|----------|-------|
> | Raw benchmark fine-tuning       | 26.92 | 10.00   | 0.00  | 0.00  | 0.00  | 0.00     | 26.09 | 20.00 | 0.00  | 47.50 | 1.33     | 39.50 |
> | Distilled benchmark fine-tuning | **56.73** | 28.50   | 46.50 | 12.00 | 32.50 | 22.75    | **74.58** | 68.50 | **43.00** | **54.50** | 36.72    | 35.00 |
> | Direct Top-3 Ensemble           | 48.80 | 25.82   | 42.45 | 12.55 | 27.68 | 19.36    | 68.98 | 67.32 | 32.43 | 19.75 | 36.89    | 31.78 |
> | Expert Fusion (naive averaging) | 55.80 | 27.67   | 39.50 | 13.10 | 31.82 | 21.74    | 69.03 | 65.00 | 31.82 | 22.60 | 32.66    | 28.55 |
> | GENOME (5-seed avg.)            | 56.66 | 27.77   | 49.34 | 15.72 | 37.19 | 22.78    | 70.69 | 68.22 | 40.00 | 35.28 | 34.97    | **43.60** |
> | GENOME+ (5-seed avg.)           | 56.44 | **30.98**   | **51.24** | **16.41** | **39.55** | **23.38**    | 74.38 | **69.89** | 41.10 | 47.06 | 38.81    | 43.54 |
>
> 1. Raw fine-tuning is clearly the worst. Direct LoRA fine-tuning on the raw benchmark pairs performs the worst overall, and even collapses to 0.00 on several datasets, consistent with our earlier claim that using only (input, final answer) pairs without reasoning traces and with noisy formats provides a very weak supervision signal.
> 2.	Even with GPT-5 distillation, gains over GENOME+ are limited. The distilled fine-tuning baseline benefits from an extra powerful teacher (GPT-5) that supplies reasoning and cleans formats, yet it only surpasses GENOME+ on a few datasets, while GENOME+ never uses any external model supervision.
> 3.	Direct aggregation is unstable. The Direct Top-3 Ensemble and naive Expert Fusion baselines exhibit highly task-dependent performance: they help on some benchmarks but degrade sharply on others, illustrating that naive aggregation of experts is unstable compared to the evolutionary GENOME/GENOME+ approach.

---

> ### Author Response · Authors · 2025-11-18
> **Q: How does this work relate and contrast to CycleQD ([1], Kuroki, et al., 2024) especially with regards to the multi-task setup?**
>
> Intuitively, our multi-task setting treats all tasks as sharing a single, unified goal: every model is scored on each task and we simply average these scores into one fitness value for evolution. In contrast, CycleQD treats different tasks as genuinely different agent problems, preserving task-specific behaviors and diversity through separate archives and then combining the best task specialists at the end.

---

> ### Author Response · Authors · 2025-11-22
> **About Data Robustness**
>
> We thank the reviewer for raising this point about robustness to validation set choice and size.
>
> First, our use of a relatively small validation set for evolution (200 examples per task) follows the standard practice in adapter-based model-merging work (e.g., LoRAHub [1], Model Swarms [2]), where a small held-out set is used to guide search/merging and a separate, larger test set is used for final reporting. In our setup (see Table 7 in the paper), for each task:
> - the validation set is randomly sampled from the benchmark and used only for evolution/selection;
> - the test set is strictly disjoint and larger than the validation set, and all numbers in the main tables are reported on this test set.
>
> To directly address the reviewer’s concern, we ran additional experiments where we vary the validation set size and resample the validation set while keeping the test set fixed. For three representative tasks (**MMLUPro, MATH, DROP**), we run GENOME and GENOME+ with validation sizes
> $ N \in \{20, 50, 100, 150, 200\} $ , and report mean ± std over 5 seeds:
>
> | MMLUPro  | N=20        | N=50        | N=100       | N=150       | N=200       |
> |----------|-------------|-------------|-------------|-------------|-------------|
> | GENOME   | 26.27±1.72  | 26.57±2.52  | 27.50±1.74  | 27.97±1.75  | 27.77±2.20  |
> | GENOME+  | 29.97±1.26  | 30.09±1.10  | 30.42±0.69  | 30.74±1.02  | 30.98±1.01  |
>
> | MATH     | N=20        | N=50        | N=100       | N=150       | N=200       |
> |----------|-------------|-------------|-------------|-------------|-------------|
> | GENOME   | 15.83±1.32  | 16.13±0.90  | 15.63±1.45  | 15.07±1.07  | 15.72±1.66  |
> | GENOME+  | 15.38±0.88  | 15.83±1.70  | 16.48±1.35  | 17.67±0.73  | 16.41±1.32  |
>
> | DROP     | N=20        | N=50        | N=100       | N=150       | N=200       |
> |----------|-------------|-------------|-------------|-------------|-------------|
> | GENOME   | 33.75±6.75  | 34.27±2.41  | 35.03±7.23  | 33.70±3.76  | 35.28±6.31  |
> | GENOME+  | 45.73±5.75  | 47.97±5.28  | 45.03±5.73  | 45.40±5.15  | 47.06±5.37  |
>
> The N=200 columns are directly copied from our main experiments in the paper.
> We can get two observations:
>
> 1.	When the validation set is very small (e.g., N=20), performance becomes somewhat less stable, which is expected.
>
> 2.	As N increases, both GENOME and GENOME+ show only mild variation, and GENOME+ consistently outperforms GENOME across all N.
>
> Overall, we see no evidence of overfitting to a particular 200-example validation split: the final test-set performance is robust to both the size and the random choice of the validation set. We will include these robustness results in the appendix of the revised manuscript.
>
> [1] Huang C, Liu Q, Lin B Y, et al. Lorahub: Efficient cross-task generalization via dynamic lora composition[J]. arXiv preprint arXiv:2307.13269, 2023.
>
> [2] Feng S, Wang Z, Wang Y, et al. Model swarms: Collaborative search to adapt llm experts via swarm intelligence[J]. arXiv preprint arXiv:2410.11163, 2024.

---

> ### Author Response · Authors · 2025-11-23
> **Can you please extend the analysis to one more large base model (>10B)?**
>
> Thank you for this suggestion. Following the main experimental setup in the paper, we extended our analysis to a larger base model, Qwen2.5-14B-Instruct, and fine-tuned 10 LoRA experts on the same set of four tasks (MMLUPro, MATH, DROP and MGSM). We repeated all dynamic methods (LoRAHub, Pack of LLMs, Model Swarms, GENOME, GENOME+) with 5 random seeds and report mean ± standard deviation. For the static baselines (Best Single, Data Merge, Expert Fusion), randomness is minimal (no evolutionary search), so we only report mean performance.
>
> | Method        | MMLUPro        | MATH           | DROP           | MGSM           |
> |--------------|----------------|----------------|----------------|----------------|
> | Best Single  | 58.74          | 36.60          | 61.20          | 65.49          |
> | Data Merge   | 53.50          | 35.50          | 19.80          | 59.54          |
> | Expert Fusion| 59.14          | 34.30          | 54.30          | 53.20          |
> | LoRAHub      | 58.27±0.99     | 39.20±0.46     | 66.27±0.23     | 64.95±0.66     |
> | Pack of LLMs | 58.44±1.14     | 28.60±0.66     | 42.33±0.45     | 51.02±0.99     |
> | Model Swarms | 60.24±0.89     | 42.00±0.50     | 64.87±1.04     | 66.64±0.79     |
> | GENOME       | 61.02±1.04     | 42.23±0.86     | 66.60±1.22     | 66.77±0.30     |
> | GENOME+      | **64.09±0.44** | **43.20±2.03** | **79.20±1.47** | **67.48±0.09** |
>
> Across all four tasks, GENOME+ remains the strongest method on Qwen2.5-14B-Instruct, with clear margins over both static baselines and other dynamic merging methods, while GENOME is consistently competitive with Model Swarms. The standard deviations are small, indicating that the gains of GENOME/GENOME+ are robust across seeds even in the >10B setting. We will include this new table and a brief discussion in the appendix of the revised manuscript.
>
> Regarding Llama-3.1-8B-Instruct, the omission from the main text was purely due to space constraints. In the original submission we prioritized showing the full spectrum of settings (single-task, multi-task, cross-task, and population scaling) on one base model in the main figures/tables, and moved the 8B results to the appendix to avoid overloading the core narrative. In the revised version, we will clearly cross-reference the corresponding appendix table, so that the role of the second base model is more visible.

---

> ### Author Response · Authors · 2025-11-23
> **About hyperparameter tuning**
>
> Regarding the comment on hyperparameter tuning, we would like to clarify that this was already analyzed in detail in Appendix A.4 of the original submission. In short, we conducted a 500-run hyperparameter study on MMLUPro, where the crossover rate (cr), individual mutation rate (imr), and gene mutation rate (gmr) were randomly sampled from 0.1, 1.0 (discretized with step 0.1), and we measured test performance, Top-3 ensemble performance, and optimization time under a fixed setup (population size N=10, $\sigma $=0.001, fixed seed and hardware). The results show that GENOME (and by extension GENOME+) is robust over a wide range of these evolutionary hyperparameters, so in our main experiments we simply adopt a single configuration from this stable region and reuse it across all tasks and base models, without additional per-task tuning. For all other baselines, we directly use the recommended hyperparameters from their original papers or public code, without granting GENOME/GENOME+ a larger hyperparameter search budget.

---

> ### Author Response · Authors · 2025-11-26
>
> We sincerely thank the reviewer's comments and suggestions. In response to your concerns, we have: (i) added robustness experiments with varying validation set sizes and resampled splits, (ii) included comparisons against full fine-tuning and direct aggregation baselines, and (iii) extended our study to a larger 14B base model. These new results have been incorporated into the revised manuscript and detailed in the appendix (highlighted in blue).
>
> We would greatly appreciate it if you could let us know whether these revisions satisfactorily address your key concerns; if not, we are very happy to further clarify or refine the experiments. If they do, we kindly ask you to consider updating your overall score in light of the empirical evidence.

---

### Author Response · Authors · 2025-11-23
**(1/2) Summary of Revisions and New Experiments for GENOME(+)**

Below is a summary of key concerns from the three reviewers and how we addressed them.

---

1. Scope, novelty, and relation to full-parameter evolutionary merging

R2 and R3 questioned whether our novelty is overstated relative to existing evolutionary model-merging work (e.g., MERGE³). In our responses “(1/2) About Weakness” and “About MERGE³ baseline” we:
- Explicitly position GENOME(+) as an adapter-based (LoRA) instantiation of evolutionary model merging, operating directly in LoRA weight space with fine-grained crossover, mutation, succession, and top-k ensembling.
- Clarify that our baseline choices follow the standard practice of prior LoRA-based works such as LoRAHub, Pack of LLMs, and Model Swarms, which compare within the adapter-based setting rather than against full-parameter methods; recent surveys also distinguish full-parameter and adapter-based merging as separate technical lines, so our focus on LoRA is consistent with existing setups rather than a special case.

Some reviewers therefore felt that MERGE³ and GENOME might be conceptually similar, calling our contribution into question. To clarify the distinction, from a mathematical perspective we note that MERGE³ optimizes over a low-dimensional recipe space $\Lambda \subset \mathbb{R}^p$ (with $p \ll d$, where $d$ is the full model parameter dimension) via a fixed merge map $\Phi : \Lambda \to \mathbb{R}^d$, $w_{\tilde m} = \Phi(\lambda)$, so the reachable set $\mathcal{H}_{\text{MERGE}^3} = \Phi(\Lambda)$ lies on at most a $p$-dimensional surface inside the $d$-dimensional parameter space. In contrast, in our adapter-based setting, GENOME defines crossover, mutation, and (in GENOME+) succession directly in the full LoRA weight space, with Gaussian perturbations acting in all adapter dimensions, so its reachable set is not confined to such a low-dimensional surface. Even at the search-space level, MERGE³ and GENOME therefore operate on fundamentally different regimes.

---

2. Baselines: from full-parameter methods to adapter-level baselines

Several concerns focused on missing or incomplete baselines.
- At the full-parameter evolutionary level, R2 and R3 asked for comparisons to MERGE³. In response, we implemented a LoRA-based variant of MERGE³, which we call MERGE3-LoRA, using the same optimization stack as the original code. We ran MERGE3-LoRA on all 12 datasets with exactly the same splits, 5 seeds, and hardware as other methods, and report mean ± std. MERGE3-LoRA is consistently below GENOME and GENOME+ on our 12-dataset suite, confirming that this finer-grained adapter-level evolution is beneficial under the LoRA setting.

- At the adapter level, R1 requested direct baselines for (i) LoRA gradient fine-tuning on the same 200 validation samples, (ii) a top-k ensemble of original experts, and (iii) naive averaging of top experts. In the response “Q: Can you provide direct baselines…” we added:
  - raw benchmark LoRA fine-tuning on (question, answer) pairs,
  - distilled benchmark LoRA fine-tuning using GPT-5 to add reasoning and normalize formats,
  - a direct Top-3 ensemble of original experts, and
  - Expert Fusion, which exactly corresponds to naive averaging of experts.

On the full 12-dataset suite, raw fine-tuning is clearly worst; distilled fine-tuning improves but generally does not surpass GENOME+; and both ensemble-only and naive-averaging baselines are unstable across tasks. All are evaluated under the same prompts and decoding setup as GENOME(+).

---

> ### Author Response · Authors · 2025-12-02
> **(2/2) Summary of Revisions and New Experiments for GENOME(+)**
>
> 3. Robustness, variance, and hyperparameters
>
> R2 and R3 stressed the need for variance/statistical information. In “About absence of standard deviations” we report, for all main methods and all 12 datasets, mean ± standard deviation over 5 fixed seeds. GENOME+ is typically the most stable method; GENOME and Model Swarms have similar variance, and LoRAHub/Pack of LLMs tend to be noisier.
>
> R1 was concerned about possible overfitting to the small 200-example validation set. In “About Data Robustness” we vary the validation size N \in \{20, 50, 100, 150, 200\} for three representative tasks (MMLUPro, MATH, DROP), resampling the validation set for each of 5 seeds while keeping a fixed disjoint test set. Performance is somewhat less stable at N=20, but otherwise varies only mildly with N, and GENOME+ consistently outperforms GENOME for all N, suggesting no overfitting to a particular 200-sample split.
>
> Regarding hyperparameters, Appendix A.4 already reports a 500-run hyperparameter study on MMLUPro for GENOME (over crossover, individual mutation, and gene mutation rates). Based on this, we select a stable configuration and reuse it for all tasks and base models, without per-task tuning. For all other baselines, we follow the recommended settings in their papers or public code.
>
> ---
>
> 4. Larger models and compute overhead
>
> R1 requested evidence beyond 2B/8B models. In “Can you please extend the analysis to one more large base model (>10B)?” we add experiments on Qwen2.5-14B-Instruct, with 10 LoRA experts on four tasks (MMLUPro, MATH, DROP, MGSM). GENOME+ remains the strongest method across all four tasks, with small standard deviations. We will also make the existing Llama-3.1-8B-Instruct results in the appendix more visible from the main text.
>
> R3 asked for FLOP-level estimates of GENOME+ vs GENOME. In “About Questions” we provide approximate FLOP counts under a unified Gemma-2-2B-it setup and show that both methods share essentially the same evaluation budget; we also give a simple runtime illustration showing that repeated full-parameter merging would be dominated by model load/unload cost, whereas LoRA-based merging keeps this overhead negligible.
>
> ---
>
> We hope this summary, together with the linked author responses, helps everyone quickly see how the paper has improved. We welcome further discussion.

---

### Note · Authors · 2026-01-05

I have read and agree with the venue's withdrawal policy on behalf of myself and my co-authors.